# RotatE: Knowledge Graph Embedding by Relational Rotation in Complex Space

**Zhiqing Sun** [1]*, **Zhi-Hong Deng**[1], **Jian-Yun Nie**[3], **Jian Tang**[2,4,5]
[1]Peking University, China
[2]Mila-Quebec Institute for Learning Algorithms, Canada
[3]Université de Montréal, Canada
[4]HEC Montréal, Canada
[5]CIFAR AI Research Chair
{1500012783, zhdeng}@pku.edu.cn
nie@iro.umontreal.ca
jian.tang@hec.ca

## Abstract

We study the problem of learning representations of entities and relations in knowledge graphs for predicting missing links. The success of such a task heavily relies on the ability of modeling and inferring the patterns of (or between) the relations. In this paper, we present a new approach for knowledge graph embedding called RotatE, which is able to model and infer various relation patterns including: symmetry/antisymmetry, inversion, and composition. Specifically, the RotatE model defines each relation as a rotation from the source entity to the target entity in the complex vector space. In addition, we propose a novel self-adversarial negative sampling technique for efficiently and effectively training the RotatE model. Experimental results on multiple benchmark knowledge graphs show that the proposed RotatE model is not only scalable, but also able to infer and model various relation patterns and significantly outperform existing state-of-the-art models for link prediction.

## 1 Introduction

Knowledge graphs are collections of factual triplets, where each triplet $(\mathtt{h}, \mathtt{r}, \mathtt{t})$ represents a relation $\mathtt{r}$ between a head entity $\mathtt{h}$ and a tail entity $\mathtt{t}$. Examples of real-world knowledge graphs include Freebase (Bollacker et al., 2008), Yago (Suchanek et al., 2007), and WordNet (Miller, 1995). Knowledge graphs are potentially useful to a variety of applications such as question-answering (Hao et al., 2017), information retrieval (Xiong et al., 2017), recommender systems (Zhang et al., 2016), and natural language processing (Yang & Mitchell, 2017). Research on knowledge graphs is attracting growing interests in both academia and industry communities.

Since knowledge graphs are usually incomplete, a fundamental problem for knowledge graph is predicting the missing links. Recently, extensive studies have been done on learning low-dimensional representations of entities and relations for missing link prediction (a.k.a., knowledge graph embedding) (Bordes et al., 2013; Trouillon et al., 2016; Dettmers et al., 2017). These methods have been shown to be scalable and effective. The general intuition of these methods is to model and infer the connectivity patterns in knowledge graphs according to the observed knowledge facts. For example, some relations are symmetric (e.g., marriage) while others are antisymmetric (e.g., filiation); some relations are the inverse of other relations (e.g., hypernym and hyponym); and some relations may be composed by others (e.g., my mother's husband is my father). It is critical to find ways to model and infer these patterns, i.e., **symmetry/antisymmetry**, **inversion**, and **composition**, from the observed facts in order to predict missing links.

Indeed, many existing approaches have been trying to either implicitly or explicitly model one or a few of the above relation patterns (Bordes et al., 2013; Wang et al., 2014; Lin et al., 2015b;

---

*This work was done when the first author was visiting Mila and Université de Montréal.

| Model | Score Function | |
|---|---|---|
| SE (Bordes et al., 2011) | $-\|\boldsymbol{W}_{r,1}\mathbf{h} - \boldsymbol{W}_{r,2}\mathbf{t}\|$ | $\mathbf{h}, \mathbf{t} \in \mathbb{R}^k, \boldsymbol{W}_{r,\cdot} \in \mathbb{R}^{k \times k}$ |
| TransE (Bordes et al., 2013) | $-\|\mathbf{h} + \mathbf{r} - \mathbf{t}\|$ | $\mathbf{h}, \mathbf{r}, \mathbf{t} \in \mathbb{R}^k$ |
| TransX | $-\|g_{r,1}(\mathbf{h}) + \mathbf{r} - g_{r,2}(\mathbf{t})\|$ | $\mathbf{h}, \mathbf{r}, \mathbf{t} \in \mathbb{R}^k$ |
| DistMult (Yang et al., 2014) | $\langle \mathbf{r}, \mathbf{h}, \mathbf{t} \rangle$ | $\mathbf{h}, \mathbf{r}, \mathbf{t} \in \mathbb{R}^k$ |
| ComplEx (Trouillon et al., 2016) | $\mathrm{Re}(\langle \mathbf{r}, \mathbf{h}, \bar{\mathbf{t}} \rangle)$ | $\mathbf{h}, \mathbf{r}, \mathbf{t} \in \mathbb{C}^k$ |
| HolE (Nickel et al., 2016) | $\langle \mathbf{r}, \mathbf{h} \otimes \mathbf{t} \rangle$ | $\mathbf{h}, \mathbf{r}, \mathbf{t} \in \mathbb{R}^k$ |
| ConvE (Dettmers et al., 2017) | $\langle \sigma(\mathrm{vec}(\sigma([\bar{\mathbf{r}}, \bar{\mathbf{h}}] * \boldsymbol{\Omega}))\boldsymbol{W}), \mathbf{t} \rangle$ | $\mathbf{h}, \mathbf{r}, \mathbf{t} \in \mathbb{R}^k$ |
| RotatE | $-\|\mathbf{h} \circ \mathbf{r} - \mathbf{t}\|^2$ | $\mathbf{h}, \mathbf{r}, \mathbf{t} \in \mathbb{C}^k, |r_i| = 1$ |

Table 1: The score functions $f_r(\mathbf{h}, \mathbf{t})$ of several knowledge graph embedding models, where $\langle \cdot \rangle$ denotes the generalized dot product, $\circ$ denotes the Hadamard product, $\otimes$ denotes circular correlation, $\sigma$ denotes activation function and $*$ denotes 2D convolution. $\bar{\phantom{x}}$ denotes conjugate for complex vectors, and 2D reshaping for real vectors in ConvE model. TransX represents a wide range of TransE's variants, such as TransH (Wang et al., 2014), TransR (Lin et al., 2015b), and STransE (Nguyen et al., 2016), where $g_{r,i}(\cdot)$ denotes a matrix multiplication with respect to relation $\mathbf{r}$.

Yang et al., 2014; Trouillon et al., 2016). For example, the TransE model (Bordes et al., 2011), which represents relations as translations, aims to model the inversion and composition patterns; the DisMult model (Yang et al., 2014), which models the three-way interactions between head entities, relations, and tail entities, aims to model the symmetry pattern. However, none of existing models is capable of modeling and inferring all the above patterns. Therefore, we are looking for an approach that is able to model and infer all the three types of relation patterns.

In this paper, we propose such an approach called RotatE for knowledge graph embedding. Our motivation is from Euler's identity $e^{i\theta} = \cos\theta + i\sin\theta$, which indicates that a unitary complex number can be regarded as a rotation in the complex plane. Specifically, the RotatE model maps the entities and relations to the complex vector space and defines each relation as a rotation from the source entity to the target entity. Given a triplet $(\mathbf{h}, \mathbf{r}, \mathbf{t})$, we expect that $\mathbf{t} = \mathbf{h} \circ \mathbf{r}$, where $\mathbf{h}, \mathbf{r}, \mathbf{t} \in \mathbb{C}^k$ are the embeddings, the modulus $|r_i| = 1$ and $\circ$ denotes the Hadamard (element-wise) product. Specifically, for each dimension in the complex space, we expect that:

$$t_i = h_i r_i, \text{ where } h_i, r_i, t_i \in \mathbb{C} \text{ and } |r_i| = 1. \tag{1}$$

It turns out that such a simple operation can effectively model all the three relation patterns: symmetric/antisymmetric, inversion, and composition. For example, a relation $\mathbf{r}$ is symmetric if and only if each element of its embedding $\mathbf{r}$, i.e. $r_i$, satisfies $r_i = e^{0/i\pi} = \pm 1$; two relations $\mathbf{r}_1$ and $\mathbf{r}_2$ are inverse if and only if their embeddings are conjugates: $\mathbf{r}_2 = \bar{\mathbf{r}}_1$; a relation $\mathbf{r}_3 = e^{i\boldsymbol{\theta_3}}$ is a combination of other two relations $\mathbf{r}_1 = e^{i\boldsymbol{\theta_1}}$ and $\mathbf{r}_2 = e^{i\boldsymbol{\theta_2}}$ if and only if $\mathbf{r}_3 = \mathbf{r}_1 \circ \mathbf{r}_2$ (i.e. $\boldsymbol{\theta_3} = \boldsymbol{\theta_1} + \boldsymbol{\theta_2}$). Moreover, the RotatE model is scalable to large knowledge graphs as it remains linear in both time and memory.

To effectively optimizing the RotatE, we further propose a novel self-adversarial negative sampling technique, which generates negative samples according to the current entity and relation embeddings. The proposed technique is very general and can be applied to many existing knowledge graph embedding models. We evaluate the RotatE on four large knowledge graph benchmark datasets including FB15k (Bordes et al., 2013), WN18 (Bordes et al., 2013), FB15k-237 (Toutanova & Chen, 2015) and WN18RR (Dettmers et al., 2017). Experimental results show that the RotatE model significantly outperforms existing state-of-the-art approaches. In addition, RotatE also outperforms state-of-the-art models on Countries (Bouchard et al., 2015), a benchmark explicitly designed for composition pattern inference and modeling. To the best of our knowledge, RotatE is the first model that achieves state-of-the-art performance on all the benchmarks.[1]

## 2 RELATED WORK

---

[1]The codes of our paper are available online: https://github.com/DeepGraphLearning/KnowledgeGraphEmbedding.

[2]The $p$-norm of a complex vector $\mathbf{v}$ is defined as $\|\mathbf{v}\|_p = \sqrt[p]{\sum |\mathbf{v}_i|^p}$. We use L1-norm for all distance-based models in this paper and drop the subscript of $\|\cdot\|_1$ for brevity.

| Model | Score Function | Symmetry | Antisymmetry | Inversion | Composition |
|:---:|:---:|:---:|:---:|:---:|:---:|
| SE | $-\|\boldsymbol{W}_{r,1}\mathbf{h} - \boldsymbol{W}_{r,2}\mathbf{t}\|$ | ✗ | ✗ | ✗ | ✗ |
| TransE | $-\|\mathbf{h} + \mathbf{r} - \mathbf{t}\|$ | ✗ | ✓ | ✓ | ✓ |
| TransX | $-\|g_{r,1}(\mathbf{h}) + \mathbf{r} - g_{r,2}(\mathbf{t})\|$ | ✓ | ✓ | ✗ | ✗ |
| DistMult | $\langle \mathbf{h}, \mathbf{r}, \mathbf{t} \rangle$ | ✓ | ✗ | ✗ | ✗ |
| ComplEx | $\text{Re}(\langle \mathbf{h}, \mathbf{r}, \bar{\mathbf{t}} \rangle)$ | ✓ | ✓ | ✓ | ✗ |
| RotatE | $-\|\mathbf{h} \circ \mathbf{r} - \mathbf{t}\|$ | ✓ | ✓ | ✓ | ✓ |

Table 2: The pattern modeling and inference abilities of several models.

Predicting missing links with **knowledge graph embedding** (KGE) methods has been extensively investigated in recent years. The general methodology is to define a score function for the triplets. Formally, let $\mathcal{E}$ denote the set of entities and $\mathcal{R}$ denote the set of relations, then a knowledge graph is a collection of factual triplets $(\mathtt{h}, \mathtt{r}, \mathtt{t})$, where $\mathtt{h}, \mathtt{t} \in \mathcal{E}$ and $\mathtt{r} \in \mathcal{R}$. Since entity embeddings are usually represented as vectors, the score function usually takes the form $f_r(\mathbf{h}, \mathbf{t})$, where $\mathbf{h}$ and $\mathbf{t}$ are head and tail entity embeddings. The score function $f_r(\mathbf{h}, \mathbf{t})$ measures the salience of a candidate triplet $(\mathtt{h}, \mathtt{r}, \mathtt{t})$. The goal of the optimization is usually to score true triplet $(\mathtt{h}, \mathtt{r}, \mathtt{t})$ higher than the corrupted false triplets $(\mathtt{h}', \mathtt{r}, \mathtt{t})$ or $(\mathtt{h}, \mathtt{r}, \mathtt{t}')$. Table 1 summarizes different score functions $f_r(\mathbf{h}, \mathbf{t})$ in previous state-of-the-art methods as well as the model proposed in this paper. These models generally capture only a portion of the relation patterns. For example, TransE represents each relation as a bijection between source entities and target entities, and thus implicitly models inversion and composition of relations, but it cannot model symmetric relations; ComplEx extends DistMult by introducing complex embeddings so as to better model asymmetric relations, but it cannot infer the composition pattern. The proposed RotatE model leverages the advantages of both.

A relevant and concurrent work to our work is the TorusE (Ebisu & Ichise, 2018) model, which defines knowledge graph embedding as translations on a compact Lie group. The TorusE model can be regarded as a special case of RotatE, where the modulus of embeddings are set fixed; our RotatE is defined on the entire complex space, which has much more representation capacity. Our experiments show that this is very critical for modeling and inferring the composition patterns. Moreover, TorusE focuses on the problem of regularization in TransE while this paper focuses on modeling and inferring multiple types of relation patterns.

There are also a large body of relational approaches for modeling the relational patterns on knowledge graphs (Lao et al., 2011; Neelakantan et al., 2015; Das et al., 2016; Rocktäschel & Riedel, 2017; Yang et al., 2017). However, these approaches mainly focus on explicitly modeling the relational paths while our proposed RotatE model implicitly learns the relation patterns, which is not only much more scalable but also provides meaningful embeddings for both entities and relations.

Another related problem is how to effectively draw negative samples for training knowledge graph embeddings. This problem has been explicitly studied by Cai & Wang (2017), which proposed a generative adversarial learning framework to draw negative samples. However, such a framework requires simultaneously training the embedding model and a discrete negative sample generator, which are difficult to optimize and also computationally expensive. We propose a self-adversarial sampling scheme which only relies on the current model. It does require any additional optimization component, which make it much more efficient.

# 3    ROTATE: RELATIONAL ROTATION IN COMPLEX VECTOR SPACE

In this section, we introduce our proposed RotatE model. We first introduce three important relation patterns that are widely studied in the literature of link prediction on knowledge graphs. Afterwards, we introduce our proposed RotatE model, which defines relations as rotations in complex vector space. We also show that the RotatE model is able to model and infer all three relation patterns.

## 3.1    MODELING AND INFERRING RELATION PATTERNS

The key of link prediction in knowledge graph is to infer the connection patterns, e.g., relation patterns, with observed facts. According to the existing literature (Trouillon et al., 2016; Toutanova & Chen, 2015; Guu et al., 2015; Lin et al., 2015a), three types of relation patterns are very important

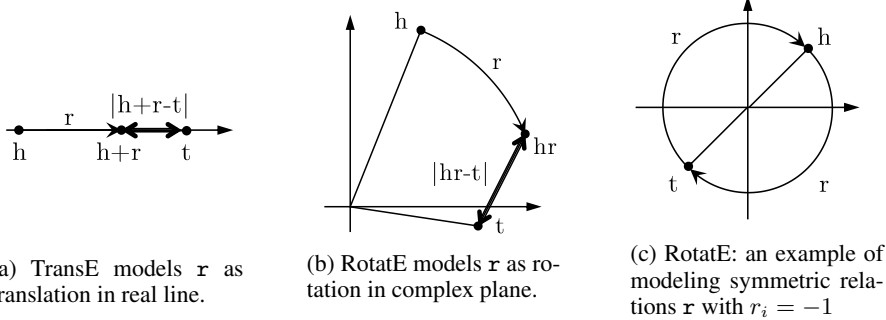

(a) TransE models r as translation in real line.

(b) RotatE models r as rotation in complex plane.

(c) RotatE: an example of modeling symmetric relations r with $r_i = -1$

Figure 1: Illustrations of TransE and RotatE with only 1 dimension of embedding.

and widely spread in knowledge graphs: symmetry, inversion and composition. We give their formal definition here:

**Definition 1.** *A relation r is **symmetric (antisymmetric)** if* $\forall x, y$

$$r(x, y) \Rightarrow r(y, x) \ ( r(x, y) \Rightarrow \neg r(y, x) )$$

*A clause with such form is a **symmetry (antisymmetry)** pattern.*

**Definition 2.** *Relation $r_1$ is **inverse** to relation $r_2$ if* $\forall x, y$

$$r_2(x, y) \Rightarrow r_1(y, x)$$

*A clause with such form is a **inversion** pattern.*

**Definition 3.** *Relation $r_1$ is **composed** of relation $r_2$ and relation $r_3$ if* $\forall x, y, z$

$$r_2(x, y) \wedge r_3(y, z) \Rightarrow r_1(x, z)$$

*A clause with such form is a **composition** pattern.*

According to the definition of the above three types of relation patterns, we provide an analysis of existing models on their abilities in inferring and modeling these patterns. Specifically, we provide an analysis on TransE, TransX, DistMult, and ComplEx.[3] We did not include the analysis on HolE and ConvE since HolE is equivalent to ComplEx (Hayashi & Shimbo, 2017), and ConvE is a black box that involves two-layer neural networks and convolution operations, which are hard to analyze. The results are summarized into Table 2. We can see that no existing approaches are capable of modeling all the three relation patterns.

## 3.2 MODELING RELATIONS AS ROTATIONS IN COMPLEX VECTOR SPACE

In this part, we introduce our proposed model that is able to model and infer all the three types of relation patterns. Inspired by Euler's identity, we map the head and tail entities h, t to the complex embeddings, i.e., $\mathbf{h}, \mathbf{t} \in \mathbb{C}^k$; then we define the functional mapping induced by each relation r as an element-wise rotation from the head entity $\mathbf{h}$ to the tail entity $\mathbf{t}$. In other words, given a triple $(\mathbf{h}, \mathbf{r}, \mathbf{t})$, we expect that:

$$\mathbf{t} = \mathbf{h} \circ \mathbf{r}, \quad \text{where } |r_i| = 1, \tag{2}$$

and $\circ$ is the Hadmard (or element-wise) product. Specifically, for each element in the embeddings, we have $t_i = h_i r_i$. Here, we constrain the modulus of each element of $\mathbf{r} \in \mathbb{C}^k$, i.e., $r_i \in \mathbb{C}$, to be $|r_i| = 1$. By doing this, $r_i$ is of the form $e^{i\theta_{r,i}}$, which corresponds to a counterclockwise rotation by $\theta_{r,i}$ radians about the origin of the complex plane, and only affects the phases of the entity embeddings in the complex vector space. We refer to the proposed model as RotatE due to its rotational nature. According to the above definition, for each triple $(\mathbf{h}, \mathbf{r}, \mathbf{t})$, we define the distance function of RotatE as:

$$d_r(\mathbf{h}, \mathbf{t}) = \|\mathbf{h} \circ \mathbf{r} - \mathbf{t}\| \tag{3}$$

By defining each relation as a rotation in the complex vector spaces, RotatE can model and infer all the three types of relation patterns introduced above. Formally, we have following results[4]:

---

[3]See discussion at Appendix A

[4]We relegate all proofs to the appendix.

| Dataset | #entity | #relation | #training | #validation | #test |
|---------|---------|-----------|-----------|-------------|-------|
| FB15k | 14,951 | 1,345 | 483,142 | 50,000 | 59,071 |
| WN18 | 40,943 | 18 | 141,442 | 5,000 | 5,000 |
| FB15k-237 | 14,541 | 237 | 272,115 | 17,535 | 20,466 |
| WN18RR | 40,943 | 11 | 86,835 | 3,034 | 3,134 |

Table 3: Number of entities, relations, and observed triples in each split for four benchmarks.

**Lemma 1.** *RotatE can infer the symmetry/antisymmetry pattern. (See proof in Appendix B)*

**Lemma 2.** *RotatE can infer the inversion pattern. (See proof in Appendix C)*

**Lemma 3.** *RotatE can infer the composition pattern. (See proof in Appendix D)*

These results are also summarized into Table 2. We can see that the RotatE model is the only model that can model and infer all the three types of relation patterns.

**Connection to TransE.** From Table 2, we can see that TransE is able to infer and model all the other relation patterns except the symmetry pattern. The reason is that in TransE, any symmetric relation will be represented by a **0** translation vector. As a result, this will push the entities with symmetric relations to be close to each other in the embedding space. RotatE solves this problem and is a able to model and infer the symmetry pattern. An arbitrary vector **r** that satisfies $r_i = \pm 1$ can be used for representing a symmetric relation in RotatE, and thus the entities having symmetric relations can be distinguished. Different symmetric relations can be also represented with different embedding vectors. Figure 1 provides illustrations of TransE and RotatE with only 1-dimensional embedding and shows how RotatE models a symmetric relation.

### 3.3 OPTIMIZATION

Negative sampling has been proved quite effective for both learning knowledge graph embedding (Trouillon et al., 2016) and word embedding (Mikolov et al., 2013). Here we use a loss function similar to the negative sampling loss (Mikolov et al., 2013) for effectively optimizing distance-based models:

$$L = -\log \sigma(\gamma - d_r(\mathbf{h}, \mathbf{t})) - \sum_{i=1}^{n} \frac{1}{k} \log \sigma(d_r(\mathbf{h}'_i, \mathbf{t}'_i) - \gamma), \tag{4}$$

where $\gamma$ is a fixed margin, $\sigma$ is the sigmoid function, and $(\mathtt{h}'_i, \mathtt{r}, \mathtt{t}'_i)$ is the $i$-th negative triplet.

We also propose a new approach for drawing negative samples. The negative sampling loss samples the negative triplets in a uniform way. Such a uniform negative sampling suffers the problem of inefficiency since many samples are obviously false as training goes on, which does not provide any meaningful information. Therefore, we propose an approach called self-adversarial negative sampling, which samples negative triples according to the current embedding model. Specifically, we sample negative triples from the following distribution:

$$p(h'_j, r, t'_j | \{(h_i, r_i, t_i)\}) = \frac{\exp \alpha f_r(\mathbf{h}'_j, \mathbf{t}'_j)}{\sum_i \exp \alpha f_r(\mathbf{h}'_i, \mathbf{t}'_i)} \tag{5}$$

where $\alpha$ is the temperature of sampling. Moreover, since the sampling procedure may be costly, we treat the above probability as the weight of the negative sample. Therefore, the final negative sampling loss with self-adversarial training takes the following form:

$$L = -\log \sigma(\gamma - d_r(\mathbf{h}, \mathbf{t})) - \sum_{i=1}^{n} p(h'_i, r, t'_i) \log \sigma(d_r(\mathbf{h}'_i, \mathbf{t}'_i) - \gamma) \tag{6}$$

In the experiments, we will compare different approaches for negative sampling.

## 4 EXPERIMENTS

### 4.1 EXPERIMENTAL SETTING

We evaluate our proposed model on four widely used knowledge graphs. The statistics of these knowledge graphs are summarized into Table 3.

- FB15k (Bordes et al., 2013) is a subset of Freebase (Bollacker et al., 2008), a large-scale knowledge graph containing general knowledge facts. Toutanova & Chen (2015) showed that almost $81\%$ of the test triplets $(\mathtt{x}, \mathtt{r}, \mathtt{y})$ can be inferred via a directly linked triplet $(\mathtt{x}, \mathtt{r}', \mathtt{y})$ or $(\mathtt{y}, \mathtt{r}', \mathtt{x})$. Therefore, the key of link prediction on FB15k is to model and infer the **symmetry/antisymmetry** and **inversion** patterns.
- WN18 (Bordes et al., 2013) is a subset of WordNet (Miller, 1995), a database featuring lexical relations between words. This dataset also has many inverse relations. So the main relation patterns in WN18 are also **symmetry/antisymmetry** and **inversion**.
- FB15k-237 (Toutanova & Chen, 2015) is a subset of FB15k, where inverse relations are deleted. Therefore, the key of link prediction on FB15k-237 boils down to model and infer the **symmetry/antisymmetry** and **composition** patterns.
- WN18RR (Dettmers et al., 2017) is a subset of WN18. The inverse relations are deleted, and the main relation patterns are **symmetry/antisymmetry** and **composition**.

**Hyperparameter Settings.** We use Adam (Kingma & Ba, 2014) as the optimizer and fine-tune the hyperparameters on the validation dataset. The ranges of the hyperparameters for the grid search are set as follows: embedding dimension[5] $k \in \{125, 250, 500, 1000\}$, batch size $b \in \{512, 1024, 2048\}$, self-adversarial sampling temperature $\alpha \in \{0.5, 1.0\}$, and fixed margin $\gamma \in \{3, 6, 9, 12, 18, 24, 30\}$. Both the real and imaginary parts of the entity embeddings are uniformly initialized, and the phases of the relation embeddings are uniformly initialized between $0$ and $2\pi$. No regularization is used since we find that the fixed margin $\gamma$ could prevent our model from over-fitting.

**Evaluation Settings.** We evaluate the performance of link prediction in the filtered setting: we rank test triples against all other candidate triples not appearing in the training, validation, or test set, where candidates are generated by corrupting subjects or objects: $(\mathtt{h}', \mathtt{r}, \mathtt{t})$ or $(\mathtt{h}, \mathtt{r}, \mathtt{t}')$. Mean Rank (MR), Mean Reciprocal Rank (MRR) and Hits at N (H@N) are standard evaluation measures for these datasets and are evaluated in our experiments.

**Baseline.** Apart from RotatE, we propose a variant of RotatE as baseline, where the modulus of the entity embeddings are also constrained: $|h_i| = |t_i| = C$, and the distance function is thus $2C \left\| \sin \frac{\boldsymbol{\theta}_h + \boldsymbol{\theta}_r - \boldsymbol{\theta}_t}{2} \right\|$ (See Equation 17 at Appendix F for a detailed derivation). In this way, we can investigate how RotatE works without modulus information and with only phase information. We refer to the baseline as pRotatE. It is obvious to see that pRotatE can also model and infer all the three relation patterns.

### 4.2 MAIN RESULTS

We compare RotatE to several state-of-the-art models, including TransE (Bordes et al., 2013), Dist-Mult (Yang et al., 2014), ComplEx (Trouillon et al., 2016), HolE (Nickel et al., 2016), and ConvE (Dettmers et al., 2017), as well as our baseline model pRotatE, to empirically show the importance of modeling and inferring the relation patterns for the task of predicting missing links.

Table 4 summarizes our results on FB15k and WN18. We can see that RotatE outperforms all the state-of-the-art models. The performance of pRotatE and RotatE are similar on these two datasets. Table 5 summarizes our results on FB15k-237 and WN18RR, where the improvement is much more significant. The difference between RotatE and pRotatE is much larger on FB15k-237 and

---

[5]Following Trouillon et al. (2016), we treat complex number as the same as real number with regard to the embedding dimension. If the same number of dimension is used for both the real and imaginary parts of the complex number as the real number, the number of parameters for the complex embedding would be twice the number of parameters for the embeddings in the real space.

| | FB15k | | | | | WN18 | | | | |
|---|---|---|---|---|---|---|---|---|---|---|
| | MR | MRR | H@1 | H@3 | H@10 | MR | MRR | H@1 | H@3 | H@10 |
| TransE [♥] | - | .463 | .297 | .578 | .749 | - | .495 | .113 | .888 | .943 |
| DistMult [♦] | 42 | .798 | - | - | **.893** | 655 | .797 | - | - | .946 |
| HolE | - | .524 | .402 | .613 | .739 | - | .938 | .930 | .945 | .949 |
| ComplEx | - | .692 | .599 | .759 | .840 | - | .941 | .936 | .945 | .947 |
| ConvE | 51 | .657 | .558 | .723 | .831 | 374 | .943 | .935 | .946 | .956 |
| pRotatE | 43 | **.799** | **.750** | .829 | .884 | **254** | .947 | .942 | .950 | .957 |
| RotatE | **40** | .797 | .746 | **.830** | .884 | 309 | **.949** | **.944** | **.952** | **.959** |

Table 4: Results of several models evaluated on the FB15K and WN18 datasets. Results of [♥] are taken from (Nickel et al., 2016) and results of [♦] are taken from (Kadlec et al., 2017). Other results are taken from the corresponding original papers.

| | FB15k-237 | | | | | WN18RR | | | | |
|---|---|---|---|---|---|---|---|---|---|---|
| | MR | MRR | H@1 | H@3 | H@10 | MR | MRR | H@1 | H@3 | H@10 |
| TransE [♥] | 357 | .294 | - | - | .465 | 3384 | .226 | - | - | .501 |
| DistMult | 254 | .241 | .155 | .263 | .419 | 5110 | .43 | .39 | .44 | .49 |
| ComplEx | 339 | .247 | .158 | .275 | .428 | 5261 | .44 | .41 | .46 | .51 |
| ConvE | 244 | .325 | .237 | .356 | .501 | 4187 | .43 | .40 | .44 | .52 |
| pRotatE | 178 | .328 | .230 | .365 | .524 | **2923** | .462 | .417 | .479 | .552 |
| RotatE | **177** | **.338** | **.241** | **.375** | **.533** | 3340 | **.476** | **.428** | **.492** | **.571** |

Table 5: Results of several models evaluated on the FB15k-237 and WN18RR datasets. Results of [♥] are taken from (Nguyen et al., 2017). Other results are taken from (Dettmers et al., 2017).

WN18RR, where there are a lot of composition patterns. This indicates that modulus is very important for modeling and inferring the composition pattern.

Moreover, the performance of these models on different datasets is consistent with our analysis on the three relation patterns (Table 2):

- On FB15K, the main relation patterns are symmetry/antisymmetry and inversion. We can see that ComplEx performs well while TransE does not perform well since ComplEx can infer both symmetry/antisymmetry and inversion patterns while TransE cannot infer symmetry pattern. Surprisingly, DistMult achieves good performance on this dataset although it cannot model the antisymmetry and inversion patterns. The reason is that for most of the relations in FB15K, the types of head entities and tail entities are different. Although DistMult gives the same score to a true triplet $(h, r, t)$ and its opposition triplet $(t, r, h)$, $(t, r, h)$ is usually impossible to be valid since the entity type of $t$ does not match the head entity type of $h$. For example, DistMult assigns the same score to $(Obama, nationality, USA)$ and $(USA, nationality, Obama)$. But $(USA, nationality, Obama)$ can be simply predicted as false since USA cannot be the head entity of the relation nationality.

- On WN18, the main relation patterns are also symmetry/antisymmetry and inversion. As expected, ComplEx still performs very well on this dataset. However, different from the results on FB15K, the performance of DistMult significantly decreases on WN18. The reason is that DistMult cannot model antisymmetry and inversion patterns, and almost all the entities in WN18 are words and belong to the same entity type, which do not have the same problem as FB15K.

- On FB15k-237, the main relation pattern is composition. We can see that TransE performs really well while ComplEx does not perform well. The reason is that, as discussed before, TransE is able to infer the composition pattern while ComplEx cannot infer the composition pattern.

- On WN18RR, one of the main relation patterns is the symmetry pattern since almost each word has a symmetric relation in WN18RR, e.g., $also\_see$ and $similar\_to$. TransE does not well on this dataset since it is not able to model the symmetric relations.

|  | Countries (AUC-PR) | | | |
|---|---|---|---|---|
|  | DistMult | ComplEx | ConvE | RotatE |
| S1 | **1.00 ± 0.00** | 0.97 ± 0.02 | **1.00 ± 0.00** | **1.00 ± 0.00** |
| S2 | 0.72 ± 0.12 | 0.57 ± 0.10 | 0.99 ± 0.01 | **1.00 ± 0.00** |
| S3 | 0.52 ± 0.07 | 0.43 ± 0.07 | 0.86 ± 0.05 | **0.95 ± 0.00** |

Table 6: Results on the Countries datasets. Other results are taken from (Dettmers et al., 2017).

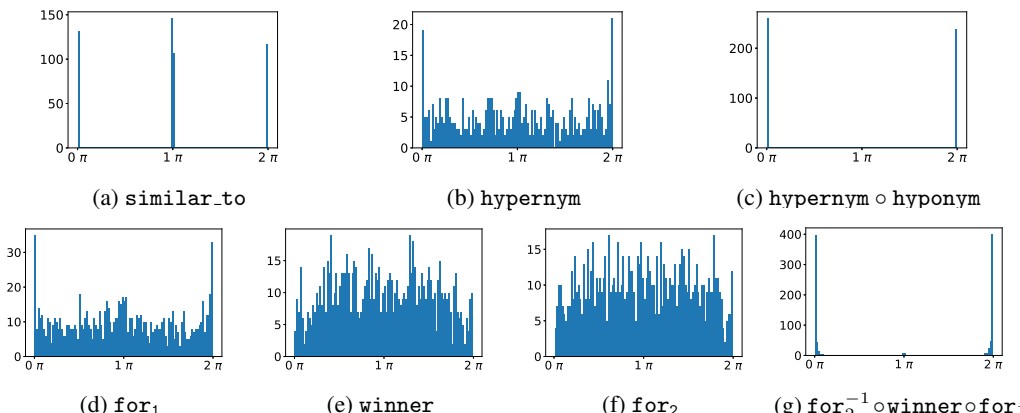

Figure 2: Histograms of relation embedding phases $\{\theta_{r,i}\}$ ($r_i = e^{i\theta_{r,i}}$), where `for`$_1$ represents relation `award_nominee/award_nominations./award/award_nomination/nominated_for`, `winner` represents relation `award_category/winners./award/award_honor/award_winner` and `for`$_2$ represents `award_category/nominees./award/award_nomination/nominated_for`. The symmetry, inversion and composition pattern is represented in Figure 2a, 2c and 2g, respectively.

## 4.3 INFERRING RELATION PATTERNS ON COUNTRIES DATASET

We also evaluate our model on the Countries dataset (Bouchard et al., 2015; Nickel et al., 2016), which is carefully designed to explicitly test the capabilities of the link prediction models for composition pattern modeling and inferring. It contains 2 relations and 272 entities (244 countries, 5 regions and 23 subregions). Unlike link prediction on general knowledge graphs, the queries in Countries are of the form `locatedIn(c, ?)`, and the answer is one of the five regions. The Countries dataset has 3 tasks, each requiring inferring a composition pattern with increasing length and difficulty. For example, task S2 requires inferring a relatively simpler composition pattern:

$$\texttt{neighborOf}(\texttt{c}_1, \texttt{c}_2) \wedge \texttt{locatedIn}(\texttt{c}_2, \texttt{r}) \Rightarrow \texttt{locatedIn}(\texttt{c}_1, \texttt{r}),$$

while task S3 requires inferring the most complex composition pattern:

$$\texttt{neighborOf}(\texttt{c}_1, \texttt{c}_2) \wedge \texttt{locatedIn}(\texttt{c}_2, \texttt{s}) \wedge \texttt{locatedIn}(\texttt{s}, \texttt{r}) \Rightarrow \texttt{locatedIn}(\texttt{c}_1, \texttt{r}).$$

In Table 6, we report the results with respect to the AUC-PR metric, which is commonly used in the literature. We can see that RotatE outperforms all the previous models. The performance of RotatE is significantly better than other methods on S3, which is the most difficult task.

## 4.4 IMPLICIT RELATION PATTERN INFERENCE

In this section, we verify whether the relation patterns are implicitly represented by RotatE relation embeddings. We ignore the specific positions in the relation embedding $\boldsymbol{\theta}_r$ and plot the histogram of the phase of each element in the relation embedding, i.e., $\{\theta_{r,i}\}$.

**Symmetry** pattern requires the symmetric relations to have property $\mathbf{r} \circ \mathbf{r} = \mathbf{1}$, and the solution is $r_i = \pm 1$. We investigate the relation embeddings from a 500-dimensional RotatE trained on WN18. Figure 2a gives the histogram of the embedding phases of a symmetric relation $similar\_to$. We can find that the embedding phases are either $\pi$ ($r_i = -1$) or $0, 2\pi$ ($r_i = 1$). It indicates that the RotatE

| | FB15k-237 | | WN18RR | | WN18 | |
|---|---|---|---|---|---|---|
| | MRR | H@10 | MRR | H@10 | MRR | H@10 |
| uniform | .242 | .422 | .186 | .459 | .433 | .915 |
| KBGAN (Cai & Wang, 2017) | .278 | .453 | .210 | .479 | .705 | **.949** |
| self-adversarial | **.298** | **.475** | **.223** | **.510** | **.736** | .947 |

Table 7: TransE with different negative sampling techniques. The results in first 2 rows are taken from (Cai & Wang, 2017), where KBGAN uses a ComplEx negative sample generator.

| | FB15k | | FB15k-237 | | Countries (AUC-ROC) | | |
|---|---|---|---|---|---|---|---|
| | MRR | H@10 | MRR | H@10 | S1 | S2 | S3 |
| TransE | .735 | .871 | .332 | .531 | $1.00 \pm 0.00$ | $1.00 \pm 0.00$ | $\mathbf{0.96 \pm 0.00}$ |
| ComplEx | .780 | **.890** | .319 | .509 | $1.00 \pm 0.00$ | $0.98 \pm 0.00$ | $0.88 \pm 0.01$ |
| RotatE | **.797** | .884 | **.338** | **.533** | $1.00 \pm 0.00$ | $1.00 \pm 0.00$ | $0.95 \pm 0.00$ |

Table 8: Results of TransE and ComplEx with self-adversarial sampling and negative sampling loss on FB15k, FB15k-237 and Countries datasets.

model does infer and model the symmetry pattern. Figure 2b is the histogram of relation `hypernym`, which shows that the embedding of a general relation does not have such a $\pm 1$ pattern.

**Inversion** pattern requires the embeddings of a pair of inverse relations to be conjugate. We use the same RotatE model trained on WN18 for an analysis. Figure 2c illustrates the element-wise addition of the embedding phases from relation $\mathbf{r}_1 = $ `hypernym` and its inversed relation $\mathbf{r}_2 = $ `hyponym`. All the additive embedding phases are 0 or $2\pi$, which represents that $\mathbf{r}_1 = \mathbf{r}_2^{-1}$. This case shows that the inversion pattern is also inferred and modeled in the RotatE model.

**Composition** pattern requires the embedding phases of the composed relation to be the addition of the other two relations. Since there is no significant composition pattern in WN18, we study the inference of the composition patterns on FB15k-237, where a 1000-dimensional RotatE is trained. Figure 2d - 2g illustrate such a $\mathbf{r}_1 = \mathbf{r}_2 \circ \mathbf{r}_3$ case, where $\theta_{2,i} + \theta_{3,i} = \theta_{1,i}$ or $\theta_{2,i} + \theta_{3,i} = \theta_{1,i} + 2\pi$.

More results of implicitly inferring basic patterns are presented in the appendix.

## 4.5 COMPARING DIFFERENT NEGATIVE SAMPLING TECHNIQUES

In this part, we compare different negative sampling techniques including uniform sampling, our proposed self-adversarial technique, and the KBGAN model (Cai & Wang, 2017), which aims to optimize a generative adversarial network to generate the negative samples. We re-implement a 50-dimension TransE model with the margin-based ranking criterion that was used in (Cai & Wang, 2017), and evaluate its performance on FB15k-237, WN18RR and WN18 with self-adversarial negative sampling. Table 7 summarizes our results. We can see that self-adversarial sampling is the most effective negative sampling technique.

## 4.6 FURTHER EXPERIMENTS ON TRANSE AND COMPLEX

One may argue that the contribution of RotatE comes from the self-adversarial negative sampling technique. In this part, we conduct further experiments on TransE and ComplEx in the same setting as RotatE to make a fair comparison among the three models. Table 8 shows the results of TransE and ComplEx trained with the self-adversarial negative sampling technique on FB15k and FB15k-237 datasets, where a large number of relations are available. In addition, we evaluate these three models on the Countries dataset, which explicitly requires inferring the composition pattern. We also provide a detailed ablation study on TransE and RotatE in the appendix.

From Table 8, we can see that similar results are observed as Table 4 and 5. The RotatE model achieves the best performance on both FB15k and FB15k-237, as it is able to model all the three relation patterns. The TransE model does not work well on the FB15k datasets, which requires modeling the symmetric pattern; the ComplEx model does not work well on FB15k-237, which requires modeling the composition pattern. The results on the Countries dataset are a little bit different, where the TransE model slightly outperforms RoateE on the S3 task. The reason is that

| Relation Category | 1-to-1 | 1-to-N | N-to-1 | N-to-N | 1-to-1 | 1-to-N | N-to-1 | N-to-N |
|---|---|---|---|---|---|---|---|---|
| **Tasks** | **Prediction Head (Hits@10)** | | | | **Prediction Tail (Hits@10)** | | | |
| TransE | .437 | .657 | .182 | .472 | .437 | .197 | .667 | .500 |
| TransH (bern) | .668 | .876 | .287 | .645 | .655 | .398 | .833 | .672 |
| KG2E_KL (bern) | .923 | .946 | .660 | .696 | .926 | .679 | .944 | .734 |
| TransE | .894 | **.972** | .567 | .880 | .879 | .671 | **.964** | .910 |
| ComplEx | **.939** | .969 | **.692** | **.893** | **.938** | **.823** | .952 | .910 |
| RotatE | .922 | .967 | .602 | **.893** | .923 | .713 | .961 | **.922** |
| **Tasks** | **Prediction Head (MRR)** | | | | **Prediction Tail (MRR)** | | | |
| TransE | .701 | .912 | .424 | .737 | .701 | .561 | .894 | .761 |
| ComplEx | .832 | .914 | **.543** | .787 | .826 | **.661** | .869 | .800 |
| RotatE | **.878** | **.934** | .465 | **.803** | **.872** | .611 | **.909** | **.832** |

Table 9: Experimental results on FB15k by relation category. The first three rows are taken from (He et al., 2015). The rest of the results are from RotatE trained with the self-adversarial negative sampling technique.

the Countries datasets do not have the symmetric relation between different regions, and all the three tasks in the Countries datasets only require inferring the region for a given city. Therefore, the TransE model does not suffer from its inability of modeling symmetric relations. For ComplEx, we can see that it does not perform well on Countries since it cannot infer the composition pattern.

### 4.7 EXPERIMENTAL RESULTS ON FB15K BY RELATION CATEGORY

We also did some further investigation on the performance of RotatE on different relation categories: one-to-many, many-to-one, and many-to-many relations[6]. The results of RotatE on different relation categories on the data set FB15k are summarized into Table 9. We also compare an additional approach KG2E_KL (He et al., 2015), which is a probabilistic framework for knowledge graph embedding methods and aims to model the uncertainties of the entities and relations in knowledge graphs with the TransE model. We also summarize the statistics of different relation categories into Table 10 in the appendix.

We can see that besides the one-to-one relation, the RotatE model also performs quite well on the non-injective relations, especially on many-to-many relations. We also notice that the probabilistic framework KG2E_KL(bern) (He et al., 2015) is quite powerful, which consistently outperforms its corresponding knowledge graph embedding model, showing the importance of modeling the uncertainties in knowledge graphs. We leave the work of modeling the uncertainties in knowledge graphs with RotatE as our future work.

## 5 CONCLUSION

We have proposed a new knowledge graph embedding method called RotatE, which represents entities as complex vectors and relations as rotations in complex vector space. In addition, we propose a novel self-adversarial negative sampling technique for efficiently and effectively training the RotatE model. Our experimental results show that the RotatE model outperforms all existing state-of-the-art models on four large-scale benchmarks. Moreover, RotatE also achieves state-of-the-art results on a benchmark that is explicitly designed for composition pattern inference and modeling. A deep investigation into RotatE relation embeddings shows that the three relation patterns are implicitly represented in the relation embeddings. In the future, we plan to evaluate the RotatE model on more datasets and leverage a probabilistic framework to model the uncertainties of entities and relations.

---

[6]Following Wang et al. (2014), for each relation r, we compute the average number of tails per head ($tphr$) and the average number of head per tail ($hptr$). If $tphr < 1.5$ and $hptr < 1.5$, r is treated as one-to-one; if $tphr \geq 1.5$ and $hptr \geq 1.5$, r is treated as a many-to-many; if $tphr < 1.5$ and $hptr \geq 1.5$, r is treated as one-to-many.

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

APPENDIX

## A  DISCUSSION ON THE ABILITY OF PATTERN MODELING AND INFERENCE

No existing models are capable of modeling all the three relation patterns. For example, TransE cannot model the symmetry pattern because it would yield $\mathbf{r} = \mathbf{0}$ for symmetric relations; TransX can infer and model the symmetry/antisymmetry pattern when $g_{r,1} = g_{r,2}$, e.g. in TransH (Wang et al., 2014), but cannot infer inversion and composition as $g_{r,1}$ and $g_{r,2}$ are invertible matrix multiplications; due to its symmetric nature, DistMult is difficult to model the asymmetric and inversion pattern; ComplEx addresses the problem of DisMult and is able to infer both the symmetry and asymmetric patterns with complex embeddings. Moreover, it can infer inversion rules because the complex conjugate of the solution to $\arg\max_{\mathbf{r}} \mathrm{Re}(\langle \mathbf{x}, \mathbf{r}, \overline{\mathbf{y}} \rangle)$ is exactly the solution to $\arg\max_{\mathbf{r}} \mathrm{Re}(\langle \mathbf{y}, \mathbf{r}, \overline{\mathbf{x}} \rangle)$. However, ComplEx cannot infer composition rules, since it does not model a bijection mapping from $\mathbf{h}$ to $\mathbf{t}$ via relation $\mathbf{r}$. These concerns are summarized in Table 2.

## B  PROOF OF LEMMA 1

*Proof.* if $\mathbf{r}(\mathbf{x}, \mathbf{y})$ and $\mathbf{r}(\mathbf{y}, \mathbf{x})$ hold, we have

$$\mathbf{y} = \mathbf{r} \circ \mathbf{x} \wedge \mathbf{x} = \mathbf{r} \circ \mathbf{y} \Rightarrow \mathbf{r} \circ \mathbf{r} = \mathbf{1}$$

Otherwise, if $\mathbf{r}(\mathbf{x}, \mathbf{y})$ and $\neg \mathbf{r}(\mathbf{y}, \mathbf{x})$ hold, we have

$$\mathbf{y} = \mathbf{r} \circ \mathbf{x} \wedge \mathbf{x} \neq \mathbf{r} \circ \mathbf{y} \Rightarrow \mathbf{r} \circ \mathbf{r} \neq \mathbf{1} \qquad \square$$

## C  PROOF OF LEMMA 2

*Proof.* if $\mathbf{r}_1(\mathbf{x}, \mathbf{y})$ and $\mathbf{r}_2(\mathbf{y}, \mathbf{x})$ hold, we have

$$\mathbf{y} = \mathbf{r}_1 \circ \mathbf{x} \wedge \mathbf{x} = \mathbf{r}_2 \circ \mathbf{y} \Rightarrow \mathbf{r}_1 = \mathbf{r}_2^{-1} \qquad \square$$

## D  PROOF OF LEMMA 3

*Proof.* if $\mathbf{r}_1(\mathbf{x}, \mathbf{z})$, $\mathbf{r}_2(\mathbf{x}, \mathbf{y})$ and $\mathbf{r}_3(\mathbf{y}, \mathbf{z})$ hold, we have

$$\mathbf{z} = \mathbf{r}_1 \circ \mathbf{x} \wedge \mathbf{y} = \mathbf{r}_2 \circ \mathbf{x} \wedge \mathbf{z} = \mathbf{r}_3 \circ \mathbf{y} \Rightarrow \mathbf{r}_1 = \mathbf{r}_2 \circ \mathbf{r}_3 \qquad \square$$

## E  PROPERTIES OF ROTATE

A useful property for RotatE is that the inverse of a relation can be easily acquired by complex conjugate. In this way, the RotatE model treats head and tail entities in a uniform way, which is potentially useful for efficient 1-N scoring (Dettmers et al., 2017):

$$\|\mathbf{h} \circ \mathbf{r} - \mathbf{t}\| = \|(\mathbf{h} \circ \mathbf{r} - \mathbf{t}) \circ \overline{\mathbf{r}}\| = \|\mathbf{t} \circ \overline{\mathbf{r}} - \mathbf{h}\| \tag{7}$$

Moreover, considering the embeddings in the polar form, i.e., $h_i = m_{h,i} e^{i\theta_{h,i}}$, $r_i = e^{i\theta_{r,i}}$, $t_i = m_{t,i} e^{i\theta_{t,i}}$, we can rewrite the RotatE distance function as:

$$\|\mathbf{h} \circ \mathbf{r} - \mathbf{t}\| = \sum_{i=1}^{k} \sqrt{(m_{h,i} - m_{t,i})^2 + 4 m_{h,i} m_{t,i} \sin^2 \frac{\theta_{h,i} + \theta_{r,i} - \theta_{t,i}}{2}} \tag{8}$$

This equation provides two interesting views of the model:

(1) When we constrain the modulus $m_{h,i} = m_{t,i} = C$, the distance function is reduced to $2C \left\| \sin \frac{\theta_h + \theta_r - \theta_t}{2} \right\|$. We can see that this is very similar to the distance function of TransE: $\|\mathbf{h} + \mathbf{r} - \mathbf{t}\|$. Based on this intuition, we can show that:

**Theorem 4.** *RotatE can degenerate into TransE. (See proof at Appendix F)*

which indicates that RotatE is able to simulate TransE.

(2) The modulus provides the lower bound of the distance function, which is $\|\mathbf{m}_h - \mathbf{m}_t\|$.

| Relation Category | 1-to-1 | 1-to-N | N-to-1 | N-to-N |
|---|---|---|---|---|
| #relation | 326 | 308 | 388 | 323 |
| #triplet (train) | 6827 | 42509 | 70727 | 363079 |
| #triplet (test) | 832 | 5259 | 8637 | 44343 |

Table 10: Statistics of FB15k by mapping properties of relations.

| | **YAGO3-10** | | | | |
|---|---|---|---|---|---|
| | MR | MRR | H@1 | H@3 | H@10 |
| DistMult | 5926 | .34 | .24 | .38 | .54 |
| ComplEx | 6351 | .36 | .26 | .40 | .55 |
| ConvE | **1671** | .44 | .35 | .49 | .62 |
| RotatE | 1767 | **.495** | **.402** | **.550** | **.670** |

Table 11: Results of several models evaluated on the YAGO3-10 datasets. Other results are taken from (Dettmers et al., 2017).

## F    PROOF OF THEOREM 4

*Proof.* By further restricting $|h_i| = |t_i| = C$, we can rewrite $\mathbf{h}, \mathbf{r}, \mathbf{t}$ by

$$\mathbf{h} = Ce^{i\boldsymbol{\theta}_h} = C\cos\boldsymbol{\theta}_h + iC\sin\boldsymbol{\theta}_h \tag{9}$$

$$\mathbf{r} = e^{i\boldsymbol{\theta}_r} = \cos\boldsymbol{\theta}_r + i\sin\boldsymbol{\theta}_r \tag{10}$$

$$\mathbf{t} = Ce^{i\boldsymbol{\theta}_t} = C\cos\boldsymbol{\theta}_t + iC\sin\boldsymbol{\theta}_t \tag{11}$$

$$\tag{12}$$

Therefore, we have

$$\|\mathbf{h} \circ \mathbf{r} - \mathbf{t}\| = C \left\| e^{i(\boldsymbol{\theta}_h + \boldsymbol{\theta}_r)} - e^{i\boldsymbol{\theta}_t} \right\| = C \left\| e^{i(\boldsymbol{\theta}_h + \boldsymbol{\theta}_r - \boldsymbol{\theta}_t)} - \mathbf{1} \right\| \tag{13}$$

$$= C \left\| \cos(\boldsymbol{\theta}_h + \boldsymbol{\theta}_r - \boldsymbol{\theta}_t) - \mathbf{1} + i\sin(\boldsymbol{\theta}_h + \boldsymbol{\theta}_r - \boldsymbol{\theta}_t) \right\| \tag{14}$$

$$= C \left\| \sqrt{(\cos(\boldsymbol{\theta}_h + \boldsymbol{\theta}_r - \boldsymbol{\theta}_t) - \mathbf{1})^2 + \sin^2(\boldsymbol{\theta}_h + \boldsymbol{\theta}_r - \boldsymbol{\theta}_t)} \right\| \tag{15}$$

$$= C \left\| \sqrt{\mathbf{2} - 2\cos(\boldsymbol{\theta}_h + \boldsymbol{\theta}_r - \boldsymbol{\theta}_t)} \right\| \tag{16}$$

$$= 2C \left\| \sin\frac{\boldsymbol{\theta}_h + \boldsymbol{\theta}_r - \boldsymbol{\theta}_t}{2} \right\| \tag{17}$$

If the embedding of $(\mathbf{h}, \mathbf{r}, \mathbf{t})$ in TransE is $\mathbf{h}', \mathbf{r}', \mathbf{t}'$, let $\boldsymbol{\theta}_h = c\mathbf{h}', \boldsymbol{\theta}_r = c\mathbf{r}', \boldsymbol{\theta}_t = c\mathbf{t}'$ and $C = 1/c$, we have

$$\lim_{c \to 0} \|\mathbf{h} \circ \mathbf{r} - \mathbf{t}\| = \|\mathbf{h}' + \mathbf{r}' - \mathbf{t}'\| \qquad \square$$

## G    LINK PREDICTION ON YAGO3-10

YAGO3-10 is a subset of YAGO3 (Mahdisoltani et al., 2013), which consists of entities that have a minimum of 10 relations each. It has 123,182 entities and 37 relations. Most of the triples deal with descriptive attributes of people, such as citizenship, gender, profession and marital status.

Table 11 shows that the RotatE model also outperforms state-of-the-art models on YAGO3-10.

## H    HYPERPARAMETERS

We list the best hyperparameter setting of RotatE w.r.t the validation dataset on several benchmarks in Table 12.

| Benchmark | embedding dimension $k$ | batch size $b$ | negative samples $n$ | $\alpha$ | $\gamma$ |
|---|---|---|---|---|---|
| FB15k | 1000 | 2048 | 128 | 1.0 | 24 |
| WN18 | 500 | 512 | 1024 | 0.5 | 12 |
| FB15k-237 | 1000 | 1024 | 256 | 1.0 | 9 |
| WN18RR | 500 | 512 | 1024 | 0.5 | 6 |
| Countries S1 | 500 | 512 | 64 | 1.0 | 0.1 |
| Countries S2 | 500 | 512 | 64 | 1.0 | 0.1 |
| Countries S3 | 500 | 512 | 64 | 1.0 | 0.1 |
| YAGO3-10 | 500 | 1024 | 400 | 1.0 | 24 |

Table 12: The best hyperparameter setting of RotatE on several benchmarks.

| | RotatE | | | | | TransE | | | | |
|---|---|---|---|---|---|---|---|---|---|---|
| | MR | MRR | H@1 | H@3 | H@10 | MR | MRR | H@1 | H@3 | H@10 |
| | negative sampling loss | | | | | | | | | |
| w/ adv | **177** | **.338** | **.241** | **.375** | **.533** | 170 | .332 | .233 | **.372** | **.531** |
| w/o adv | 185 | .297 | .205 | .328 | .480 | 175 | .297 | .202 | .331 | .486 |
| | margin-based ranking criterion | | | | | | | | | |
| w/ adv | 225 | .322 | .225 | .358 | .516 | 167 | **.333** | **.237** | .370 | .522 |
| w/o adv | 199 | .293 | .202 | .324 | .476 | **164** | .306 | .212 | .340 | .493 |

Table 13: Results of ablation study on FB15k-237, where "adv" represents "self-adversarial".

## I  ABLATION STUDY

Table 13 shows our ablation study of self-adversarial sampling and negative sampling loss on FB15k-237. We also re-implement a 1000-dimension TransE and do ablation study on it. From the table, We can find that self-adversarial sampling boosts the performance for both models, while negative sampling loss is only effective on RotatE; in addition, our re-implementation of TransE also outperforms all the state-of-the-art models on FB15k-237.

## J  VARIANCE OF THE RESULTS

In Table 14, We provide the average and variance of the MRR results on FB15k, WN18, FB15k-237 and WN18RR. Both the average and the variance is calculated by three runs of RotatE with difference random seeds. We can find that the performance of RotatE is quite stable for different random initialization.

## K  MORE RESULTS OF IMPLICIT BASIC PATTERN INFERENCE

We provide more histograms of embedding phases in Figure 3 - 5.

|       | FB15k          | WN18           | FB15k-237      | WN18RR         |
|-------|----------------|----------------|----------------|----------------|
| MRR   | $.797 \pm .001$ | $.949 \pm .000$ | $.337 \pm .001$ | $.477 \pm .001$ |

Table 14: The average and variance of the MRR results of RotatE on FB15k, WN18, FB15k-237 and WN18RR.

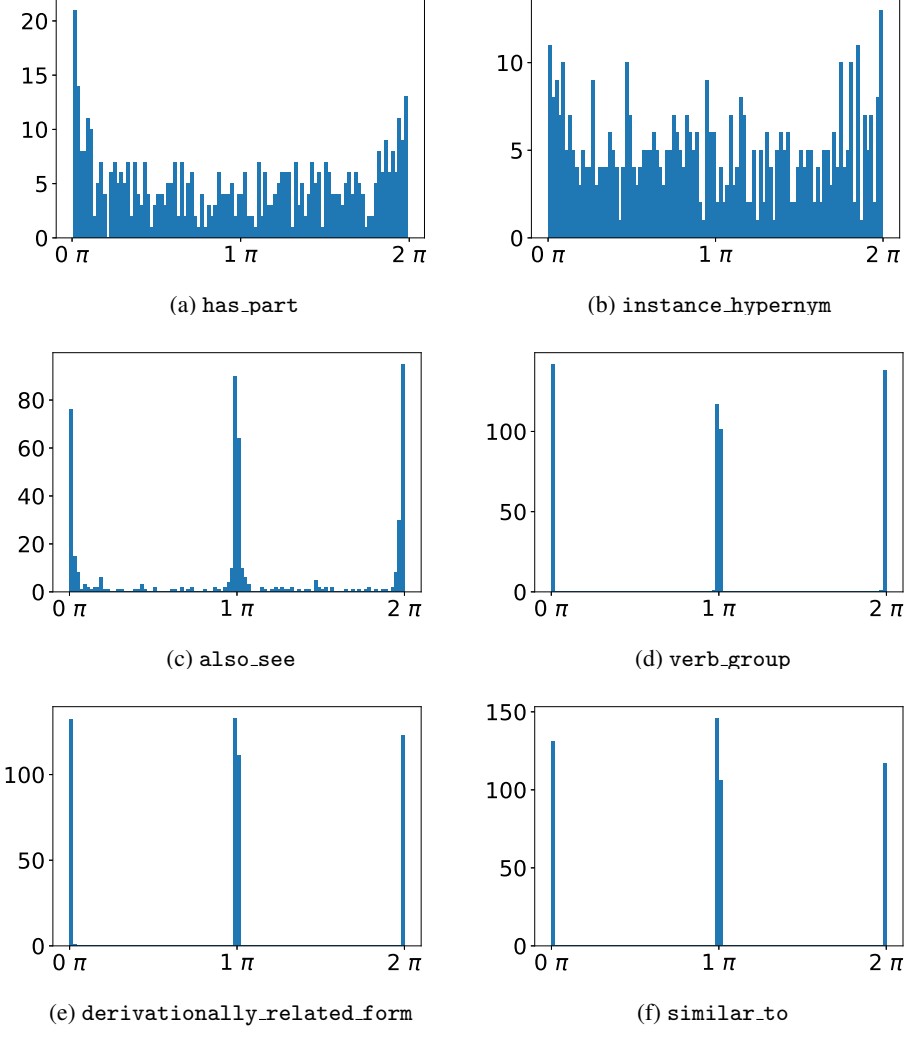

Figure 3: Histograms of embedding phases from two general relations and four symmetric relations on WN18. ( $k = 500$ )

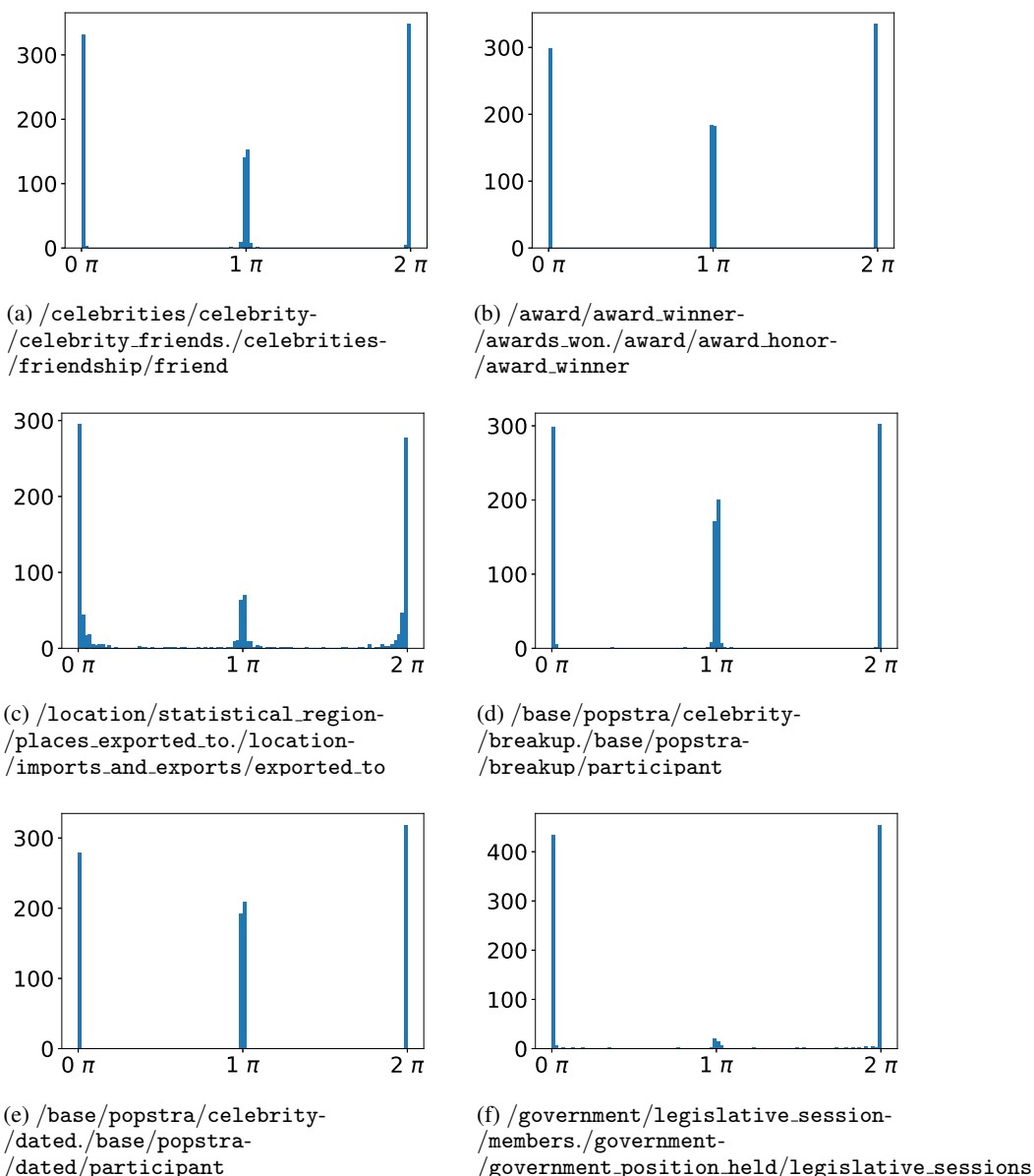

(a) /celebrities/celebrity-
/celebrity_friends./celebrities-
/friendship/friend

(b) /award/award_winner-
/awards_won./award/award_honor-
/award_winner

(c) /location/statistical_region-
/places_exported_to./location-
/imports_and_exports/exported_to

(d) /base/popstra/celebrity-
/breakup./base/popstra-
/breakup/participant

(e) /base/popstra/celebrity-
/dated./base/popstra-
/dated/participant

(f) /government/legislative_session-
/members./government-
/government_position_held/legislative_sessions

Figure 4: Histograms of embedding phases from six symmetric relations on FB15k-237. ($k = 1000$)

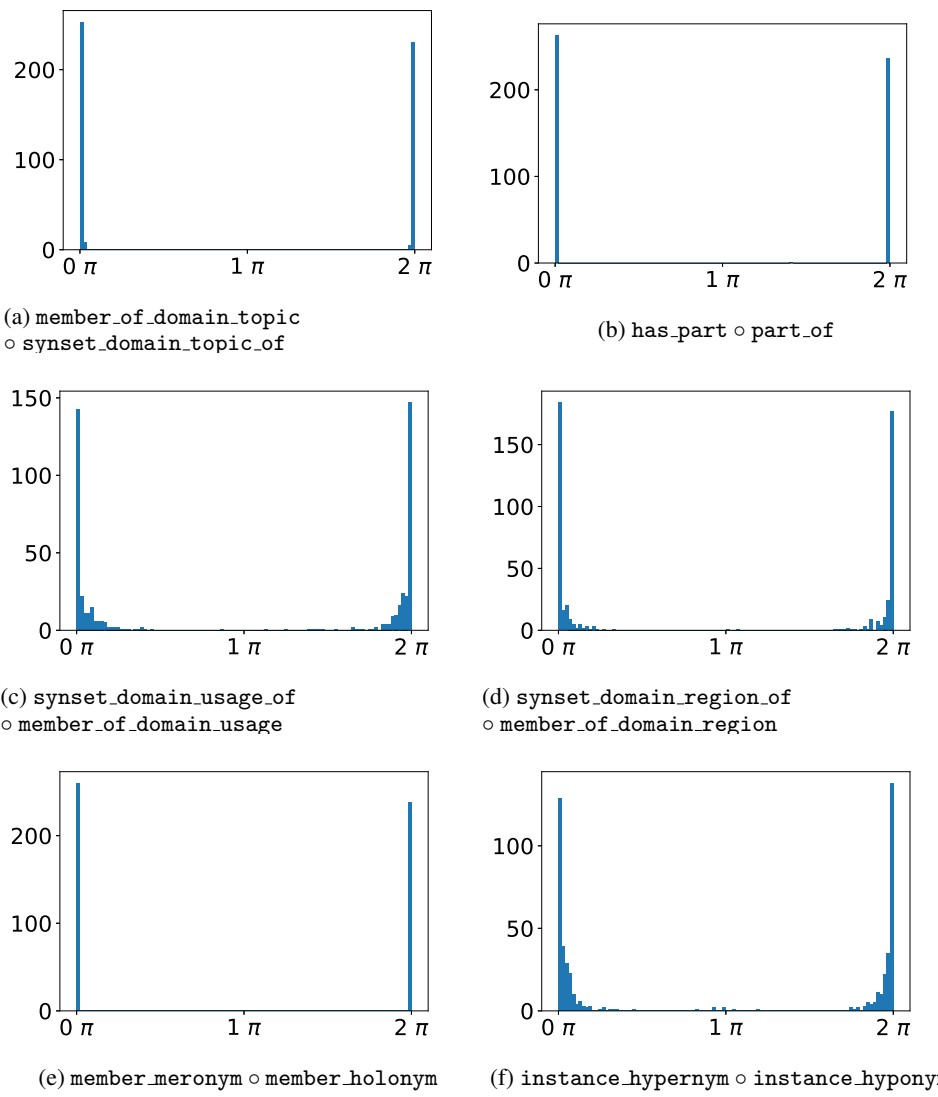

Figure 5: Histograms of element-wise additions of inversed relation embedding phases on WN18. ($k = 500$)

