# OpenReview forum: "RotatE: Knowledge Graph Embedding by Relational Rotation in Complex Space"
_ICLR.cc/2019/Conference_

### Official Review · AnonReviewer2 · 2018-10-22
**Is it the RotatE scoring function or the adversarial sampling?**

**Rating:** 7
**Confidence:** 4

**Review:**

# Summary
This paper presents a neural link prediction scoring function that can infer symmetry, anti-symmetry, inversion and composition patterns of relations in a knowledge base, whereas previous methods were only able to support a subset. The method achieves state of the art on FB15k-237, WN18RR and Countries benchmark knowledge bases. I think this will be interesting to the ICLR community. I particularly enjoyed the analysis of existing methods regarding the expressiveness of relational patterns mentioned above.

# Strengths
- Improvements over prior neural link prediction methods
- Clearly written paper
- Interesting analysis of existing neural link prediction methods

# Weaknesses
- As the authors not only propose a new scoring function for neural link prediction but also an adversarial sampling mechanism for negative data, I believe a more careful ablation study should have been carried out. There is an ablation study showing the impact of the negative sampling on the baseline TransE, as well as another ablation in the appendix demonstrating the impact of negative sampling on TransE and the proposed method, RotatE, for the FB15k-237. However, from Table 10 in the appendix, one can see that the two competing methods, TransE and RotatE, in fact, perform fairly similarly once both use adversarial sampling it still remains unclear whether the gains observed in table 4 and 5 are due to adversarial sampling or a better scoring function. Particularly, I want to see results of a stronger baseline, ComplEx, equipped with the adversarial sampling approach. Ideally, I would also like to see multiple repeats of the experiments to get a sense of the variance of the results (as it has been done for Countries in Table 6).

# Minor Comments
- Eq 5: Already introduce gamma (the fixed margin) here.
- While I understand that this paper focuses on knowledge graph embeddings, I believe the large body of other relational AI approaches should be mention as some of them can also model symmetry, anti-symmetry, inversion and composition patterns of relations as well (though they might be less scalable and therefore of less practical relevance), e.g. the following come to mind:
  - Lao et al. (2011). Random walk inference and learning in a large scale knowledge base.
  - Neelakantan et al. (2015). Compositional vector space models for knowledge base completion.
  - Das et al. (2016). Chains of Reasoning over Entities, Relations, and Text using Recurrent Neural Networks.
  - Rocktaschel and Riedel (2017). End-to-end Differentiable Proving.
  - Yang et al. (2017). Differentiable Learning of Logical Rules for Knowledge Base Completion.
- Table 6: How many repeats were used for estimating the standard deviation?


Update: I thank the authors for their response and additional experiments. I am increasing my score to 7.

---

> ### Author Response · Authors · 2018-11-26
> **Thanks for the great comments and suggestions!**
>
> “Particularly, I want to see results of a stronger baseline, ComplEx, equipped with the adversarial sampling approach….”
>
> We have added the experimental results of TransE and ComplEx on three datasets in our paper (Table 8). We can see that our proposed approach still outperforms ComplEx with the new adversarial approach, especially on the data set FB15k-237 and Countries. The reason is that  FB15k-237 and Countries contain many composition patterns, which cannot be modeled by ComplEx but can be effectively modeled by RotatE.
>
> “Ideally, I would also like to see multiple repeats of the experiments to get a sense of the variance of the results...”
>
> We also added the variance of the results of our model on different data sets, which are summarized into Table 12 in the appendix. We can see that the variance of the results are very small, 0.001 at maximum.
>
> “Table 6: How many repeats were used for estimating the standard deviation?”
>
> Only 3 are used. Since the variance are very small, the same results are obtained with more repeats.
>
> “While I understand that this paper focuses on knowledge graph embeddings, I believe the large body of other relational AI approaches should be mention….”
>
> We have added some discussion on these methods in the related work section.

---

> > ### Comment · AnonReviewer2 · 2018-11-28
> > **1-to-N, N-to-1, N-to-N?**
> >
> > Thanks a lot for the response and updating the paper.
> >
> > What is your response to the public comment above?
> > https://openreview.net/forum?id=HkgEQnRqYQ&noteId=rkgeGrg_Tm
> >
> > Specifically, if TransE and RotatE suffer from not being able to model 1-to-N, N-to-1, N-to-N relations, what is your take on why this is not reflected in the experimental results for RotatE? Is this a limitation of the used datasets?
> >
> >
> > -- R2

---

> > > ### Author Response · Authors · 2018-11-28
> > > **The RotatE model is somehow able to model 1-to-N relations.**
> > >
> > > We first would like to provide some theoretical analysis to show that the RotatE model can also somehow model the 1-to-N relations. Taking a 1-to-N relation r as an example. The triplets having the head entity x and relation r are denoted as: r(x, y1), r(x, y2) …. r(x, yn). When the optimization converges, it could be easily to find out that the embeddings of y1, y2, …, yn will be evenly distributed on the surface of a hypercube (or a hypersphere in the case of L-2 norm) centered at rx. In other words, ||rx - y1|| = ||rx - y2|| = .. = ||rx - yn||. This phenomenon is the same as in semantic matching models, like ComplEx, where the scores <r,x,\bar{y1}>=<r,x,\bar{y2}>=..=<r,x,\bar{yn}>. Therefore, the RotatE model can somehow deal with 1-to-N relations just like ComplEx, as well as TransE.
> > >
> > > A more elegant and rigorous approach to model the 1-to-N, N-to-1, and N-to-N relations is to leverage a probabilistic framework to model the uncertainties of the entities, where each predicted entity is represented as a Gaussian distribution. This has been proved quite effective in [1]. Our RotatE model can easily leverage this framework to mitigate this issue.
> > >
> > > Another thing to note is that the focus of this paper is to model and infer the different types of relation patterns, but not the 1-to-N, N-to-1, and N-to-N relationships. However, we will conduct further experiments to compare the performance of different methods (TransE, ComplEx and RotatE) on the 1-1, 1-to-N, N-to-1, and N-to-N relationships.
> > >
> > > [1] Shizhu He, Kang Liu, Guoliang Ji and Jun Zhao, Learning to Represent Knowledge Graphs with Gaussian Embedding

---

### Official Review · AnonReviewer1 · 2018-11-03
**Solid work**

**Rating:** 7
**Confidence:** 4

**Review:**

The paper proposes a method for graph embedding to be used for link prediction, in which each entity is represented as a vector in complex space and each relation is modeled as a rotation from the head entity to the tale entity.
From the modeling perspective, the proposed model is rich as many type of relations can be modeled with it. In particular, symmetric and anti-symmetric relations can be modeled. It is also possible to model the inverse of a relation and the composition of two relations with this setup. Empirical evaluation demonstrates that method is effective and beats a number of well known competitors.

This is a solid work and could be of interest in the community. Modeling is elegant and experimental results are strong.
I have not seen it proposed before.

- The presentation of paper could be improved, in particular the first paragraph of page 2 where the representation in complex domain is introduced is hard to follow and could be improved by inserting formulations instead of merely text.
It would be nice to explicitly mention the number of real and imaginary dimensions of the complex vectors and provide explicit formulation for the Hadamard product on the complex domain, since the term elementwise could be ambiguous.
- The optimization section does not mention how constraints are imposed. This is an important technicality and should be clarified.
- In experiments, how does the effective number of parameters that are used to express representations compare when the representations are a complex vs a real number? Each complex number is presented with two parameters and each real number with one parameter. How is that taken into account in experiments
- Since the method is reported to beat several number of competitors, it is useful to provide the code.


Based on the results above, I vote for the paper to be accepted.

---

> ### Author Response · Authors · 2018-11-26
> **Thanks for your appreciation and great suggestions!**
>
> Thanks for your appreciation to our work and the great comments. We’ve revised the introduction part on the representations in complex domain.
>
> “The optimization section does not mention how constraints are imposed.”
>
> Since each relation is modeled as a rotation in the complex vector space, we represent each relation r according to its polar form with its modulus as 1, i.e.,
> Re(r) = sine(\theta), and Im(r) = cosine(\theta), where \theta is the phase of relation r. With the polar form representation, the constraints can be easily satisfied.
>
> “In experiments, how does the effective number of parameters that are used to express representations compare when the representations are a complex vs a real number ….”
>
> If the same number of dimension is used for both the real and imaginary parts of the complex number as the real number, the number of parameters for the complex embedding would be twice the number of parameters for the embeddings in the real space. To make a fair comparison, in the process of grid search for finding the optimum embedding dimension, we double the range of the search space  for models represented in real space such as TransE.
>
> “Since the method is reported to beat several number of competitors, it is useful to provide the code.”
>
> Yes, we will definitely release our code and share it with the entire community.

---

### Official Review · AnonReviewer3 · 2018-11-04
**This paper is an important new contribution to the field.   The results should be compared to TorusE.**

**Rating:** 7
**Confidence:** 3

**Review:**

The authors propose to model the relations as a rotation in the complex vector space. They show that this way one can model symmetry/antisymmetry, inversion and composition. Another contribution is the so-called self-adversarial negative sampling.

Pros:  The problem that they raise is important and the solution is relevant. The results considering the simplicity of the proposed model are impressive. The experiments, proof of lemmas and general overview are easy to follow, well-written and well-organized.  The improvement given the negative sampling approach is also noteworthy.

Cons: Nevertheless, this approach is very similar to TorusE [1], since the element-wise rotation on the complex plane is somehow related to transformation on high-dimensional Torus. Therefore, it is expected from the authors to investigate the differences between these two approaches.

Suggestions:
Also, it is important to note the result of ablation study on Table 10 in supplementary materials, since part of the improvement does not come only from how the authors model the relation but also from the negative sampling(which could improve the results of other works as well). Maybe it is even better if Table 10 is presented in the main paper.
Another suggestion is to mention the negative sampling contribution also in the abstract.


[1] Ebisu, Takuma, and Ryutaro Ichise. "Toruse: Knowledge graph embedding on a lie group." arXiv preprint arXiv:1711.05435 (2017)."

---

> ### Author Response · Authors · 2018-11-26
> **Thanks for your appreciation to our work and mentioning another relevant work!**
>
> Thanks for your appreciation to our work and your great comments on improving the paper. We have added the experimental results of TransE and ComplEx with self-adversarial negative sampling on three datasets in our paper (Table 8). We have also added the contribution of the self-adversarial negative sampling into both the abstract and introduction.
>
> Regarding TorusE, thanks again for bringing it to our attention, which we did not notice before. It is indeed relevant to our model, which is a concurrent work. We have discussed this model in the related work section. The difference between TorusE and RotatE can be summarized as below:
>
> (1) The TorusE model constraints the embedding of objects on a torus, and models relations as translations, while the RotatE model embeds objects on the entire complex vector space, and models relations as rotations.
>
> (2) The TorusE model requires embedding objects on a compact Lie group [2] while the RotatE model allows embedding objects on a non-compact Lie group, which has much more representation capacity. The TorusE model is actually very close to a special case of our model, i.e., pRotatE, which constraints the modulus of the head and entity embeddings fixed. As shown in Table 5, it is very important for modeling and inferring the composition patterns by embedding the entities on a non-compact Lie group. We can also compare the results of TorusE and RotatE on the FB15k and WN18 data sets (Table 3 in the TorusE paper and Table 4 in our paper), we can see that our RotatE model significantly outperforms TorusE on the two data sets.
>
> (3) The motivations of the TorusE paper and this paper are quite different. The TorusE paper aims to solve the regularization problem of TransE, while our paper focuses on inferring and modeling three important and popular relation patterns.
>
> [1] Ebisu, Takuma, and Ryutaro Ichise. "Toruse: Knowledge graph embedding on a lie group." arXiv preprint arXiv:1711.05435 (2017)."
> [2] https://en.wikipedia.org/wiki/Compact_group#Compact_Lie_groups

---

### Public Comment · (anonymous) · 2018-09-29
**Not mention results of ConvKB and Reciprocal ComplEx-N3**

You should mention the experimental results of ConvKB [1] and Reciprocal ComplEx-N3 [2]. Reciprocal ComplEx-N3 gives higher MRR and Hits@10 scores than yours on both FB15K and FB15k-237. ConvKB produces better scores than yours for MRR on FB15k-237 and MR on WN18RR.

[1] A Novel Embedding Model for Knowledge Base Completion Based on Convolutional Neural Network. NAACL-HLT 2018.
[2] Canonical Tensor Decomposition for Knowledge Base Completion. ICML-2018. Oral presentation.

---

> ### Author Response · Authors · 2018-10-29
> **Thanks for pointing this out!**
>
> Thanks for pointing this out! We’re aware of the result of ConvKB, which achieves a very high MRR on FB15k (0.396). The reason that we did compare with ConvKB [1] is that there is a bug in ConvKB’s evaluation.
>
> We tried to reproduce their results from their published code [2], but found that the ConvKB tends to assign the same score, i.e., 0,  to many triplets. The reason is that the RELU activation function is used in the convolution layers, which tends to have very sparse output, i.e., the output of many neurons are zero. This brings a big problem in the evaluation.
>
> For evaluation, given a query (h,r, ?), the goals is to identify the rank of the true positive triplets (h, r, t) among all the possible (h, r, t’) triplets. Since the scores of many triplets given by ConvKB equal to 0 (typo, should be "the same score" or "bias"), the true positive triplets and many other false triplets are all ranked the first position at the same time. A reasonable solution would be to randomly pick a triplet among those triplets as the first ranked triplet, and so on. However,  we find that a specific ranking procedure is used by ConvKB, which tends to rank the true positive triplets in a high position. As a result, the performance evaluated in this way is really high, which is not true in reality. We strongly suggest the authors of ConvKB to take a look at this issue and fix their results.
>
> For the results of Reciprocal ComplEx-N3, thanks again for pointing this out, which we are not aware of before the submission. However, note that the focus of the Reciprocal ComplEx-N3 and this paper is different. Our paper proposes a new distance function for learning knowledge graph embedding, and our proposed RotatE is able to infer three relation patterns including composition, symmetry/asymmetry, and inversion, which offers good model interpretability. The focus of Reciprocal ComplEx-N3, however, is on different regularization techniques, which could be potentially applied to our proposed RotatE model. For example, on the FB15k data set, the performance of RotatE increases from 0.797 to 0.815 with the N3-regularizer, which outperforms  the performance of ComplEx-N3 on FB15k (0.80).  We are still in the process of implementing the reciprocal setting for our RotatE model, which seems to be pretty effective according to [3].
>
> [1] A Novel Embedding Model for Knowledge Base Completion Based on Convolutional Neural Network
> [2] https://github.com/daiquocnguyen/ConvKB
> [3] Canonical Tensor Decomposition for Knowledge Base Completion

---

> > ### Public Comment · ~Dai_Quoc_Nguyen1 · 2018-10-29
> > **No bug in our ConvKB evaluation!**
> >
> > Disclose: I am the author of ConvKB. I had re-run my ConvKB implementation. And there is not a single triple having score at 0 on FB15k-237.
> >
> > It would be nice if you can create an open issue in my ConvKB github before discussing any information made in public.
> >
> > Update for a clarification:
> >
> > It is important to note that our implementation can work with other score functions. Last year, I verified my “eval.py” implementation by using the same output vector and matrix embeddings produced by other models (such as TransE, TransR, TransH and STransE) to prove that our "eval.py" implementation is correct and can produce the exact same scores as produced by those models.
> >
> > For each correct test triple, I just replicated this correct test triple several times to add to its set of corrupted triples, in order to work with a batch size (as shown in Lines 188-190 in “eval.py”). This is straightforward and does not matter when ranking the correct test triple. I thought that "the same score" you mentioned is actually for the correct test triple because of replicating. You should have a careful look at this point and then edit your comment above to have a reasonable reply.
> >
> > I just read your paper. This is nice work. Your experimental results are still great even you add negative results from other papers.

---

> > > ### Author Response · Authors · 2018-11-03
> > > **Sorry we meant that many triples have the same score, and an open evaluation code is now available.**
> > >
> > > Hi Dai,
> > >      Thanks for the verification. In the above comment, sorry we meant that many triplets have the same score, which equals to the bias of your model, i.e., b = tf.Variable(tf.constant(0.0, shape=[num_classes]), name="b") in your model.py code. The reasons is that in many cases all the nonlinear RELU units are not activated. In addition, we found that this problem would only occur when the nonlinear activation Relu is used in the model. This explains why the evaluation of other models, including TransE, TransR, TransH and STransE, are correct.
> > >
> > > We suggest you to re-evaluate your model without replicating the true triplets. We’ve fixed this debug in your code and put the updated codes in https://github.com/KnowledgeBaseCompleter/eval-ConvKB .
> > >
> > > By the way, we appreciate your work, which we find is really interesting.  We did not intend any offence to your work. We hope we can push forward this exciting direction together. We look forward to your feedback.

---

> > > > ### Public Comment · ~Dai_Quoc_Nguyen1 · 2018-11-04
> > > > **No bug in our ConvKB evaluation!**
> > > >
> > > > How many valid test triples and their corrupted triples have the same score. And what are they and their ranks on WN18RR and FB15k237? You had mentioned "equal to 0" (the same score) in your first reply. It seems that you actually did not run my code before. I do not want to discuss our model in details as my code was based on Denny Britz's implementation for employing a CNN to text classification.
> > > >
> > > > There is nothing called "a specific ranking procedure" in my evaluation. I do not know why you must pay much attention to "replicating the valid test triples". Again, this is straightforward and does not matter when ranking, because each valid test triple and its replicated triples have a same score and a same rank.
> > > >
> > > > As I said in my first reply, it would be nice if you created an open issue in my ConvKB github for further discussions. So I could tell you that we also had another version to evaluate the model without replicating the valid test triples, for which the experimental results are still same for with and without replicating the triples. This obviously helps to save time for both of us.
> > > >
> > > > The "without replicating" version ran slower than the version in the github, thus I did not update it last year. But now, I have just updated it to my ConvKB github. You can check and test it.
> > > >
> > > > Your approach and results are great. And you do not need to beat all scores on all datasets to have an accepted paper. I appreciate if you can also include our published results.

---

> > > > > ### Author Response · Authors · 2018-11-26
> > > > > **Your updated codes still have the same problem.**
> > > > >
> > > > > Thanks for your verification for your model. We do agree that the implementation of your model is correct. However, what we pointed out is that your evaluation is problematic!!
> > > > >
> > > > > For your updated eval.py, we find that you used the following code to get the rank for each triplets:
> > > > >
> > > > > results_with_id = rankdata(results, method='min')
> > > > >
> > > > > where ‘min’ represents “The minimum of the ranks that would have been assigned to all the tied values is assigned to each value. (This is also referred to as “competition” ranking.)” according to the official document [1].
> > > > >
> > > > > However, such "a specific ranking procedure" tends to rank the true positive triplets in a high position, if there are many triplets with the same score.
> > > > >
> > > > > A simple example is that a model produce score=b for all triplets, then results_with_id = rankdata(results, method='min') will return the results that all the triplets are ranked in the first position. In other words, in this case MRR = 1, which is definitely wrong.
> > > > >
> > > > > Moreover, as mentioned in [2], we have fixed the bug in your previous codes and reported the true performance of your model on FB15k-237. We provided the checkpoint file, where you can check that MRR = 40 by your original eval.py, but 24 by our bug-fixed eval.py.
> > > > >
> > > > > As for your updated codes, we suggest that you should replace the “rankdata” part by:
> > > > >
> > > > > results_with_id = rankdata(results, method=’ordinal’)
> > > > >
> > > > > where ‘ordinal’ represents “All values are given a distinct rank, corresponding to the order that the values occur in a.”  according to the official document [1]. Although the results may be a little different from the results of our released bug-fixed eval.py [2] (We used quicksort ranking by following you), it would also provide a valid evaluation for your model.
> > > > >
> > > > > For the previous codes, We opened a pull that fix the bug (https://github.com/daiquocnguyen/ConvKB/pull/3), but it was closed. For your new codes, we also opened a pull to fix the bug (https://github.com/daiquocnguyen/ConvKB/pull/4).
> > > > >
> > > > > Finally, we want to emphasize again that we did not intend any offence to your work. The truth is that we found a problem, and we want to make it right.
> > > > >
> > > > > [1]: https://docs.scipy.org/doc/scipy/reference/generated/scipy.stats.rankdata.html
> > > > > [2]: https://github.com/KnowledgeBaseCompleter/eval-ConvKB

---

> > > > > > ### Public Comment · ~Dai_Quoc_Nguyen1 · 2018-11-28
> > > > > > **Still no bug in our ConvKB evaluation!**
> > > > > >
> > > > > > 1.  " results_with_id = rankdata(results, method='min') ": I used this last year because I simply want to give the valid test triple and its replicated triples a same rank (since I used a batch size).
> > > > > >
> > > > > > 2. " A simple ... wrong": your example is not real since a model tends to give high scores to valid triples and low score to invalid triples. None of existing models can give a MRR score of 1.
> > > > > >
> > > > > > I have another question for you: Assume that a valid test triple and some of its corrupted triples have a same score. Why must you think it is wrong if assigning them a same rank?
> > > > > >
> > > > > > 3. " For the previous codes, We opened ... /pull/4). ": I keep to maintain my code and do not accept any pull request before/without opening an issue in my ConvKB github for a discussion. As I said in my previous reply, it could be much better if you created an open issue in my github with your official account.
> > > > > >
> > > > > > 4. " results_with_id = rankdata(results, method=’ordinal’) ": I just updated my code using "ordinal" and still get a same results (with a quick test using pre-trained TransE embeddings). You can check and test it for ConvKB. No bug.
> > > > > >
> > > > > > 5. I will not discuss about the implementation of my evaluation further, here. If you still have other problems, you can create an open issue in my ConvKB github.
> > > > > >
> > > > > > Thank you for your time and discussion.

---

> > > > > > > ### Comment · Area_Chair1 · 2018-11-28
> > > > > > > **Clarify why 'ordinal' is not sufficient?**
> > > > > > >
> > > > > > > This will likely not be taken into account for the decision, so I don't want to discuss this too much.
> > > > > > >
> > > > > > > But it is an important issue for the field, and I understand the concern raised by the authors: triples with the same score should get random (or ideally, max) ranking, not min. With min, the MRR ranking will be inflated, incorrectly, and benefits methods that tend to produce tied scores.
> > > > > > >
> > > > > > > I have a quick question for the authors though. Can you verify, and explain, why rankdata(results, method=’ordinal’) is not sufficient? Is it because the true triple comes earlier in the list (somehow)?

---

> > > > > > > > ### Author Response · Authors · 2018-11-29
> > > > > > > > **You are right!!**
> > > > > > > >
> > > > > > > > Thanks for your understanding! You are right! ‘ordinal’ is not sufficient in the case when the true triple comes earlier in the list, especially when the true triplet is put in the beginning of the list. The ConvKB’s updated new eval.py [1] suffers this problem by always putting the true triplet in the first position (see the codes below).
> > > > > > > >
> > > > > > > > #thus, insert the valid test triple again, to the beginning of the array
> > > > > > > > new_x_batch = np.insert(new_x_batch, 0, x_batch[i], axis=0)
> > > > > > > > new_y_batch = np.insert(new_y_batch, 0, y_batch[i], axis=0)
> > > > > > > >
> > > > > > > > In this case, ‘ordinal’ is essentially equivalent to ‘min’, so it’s not sufficient. However, this problem can be easily addressed by randomly shuffling the list.
> > > > > > > >
> > > > > > > > [1] https://github.com/daiquocnguyen/ConvKB/commit/c7ee60526ee81b46c2b0075cca2e387b0dbc6e90

---

### Public Comment · (anonymous) · 2018-11-06
**What is the difference compared to the ComplEx embedding model?**

The reported results are high, which raise my interest. But, it also raises attention to some important issues that need to be addressed.

The proposed model is very similar to the ComplEx embedding model [1]. In fact, in the ComplEx model, the score function is $ real(<r, h, \bar{t}>) $, which includes the element-wise product between $ r \circ h $. Because the ComplEx model uses complex-value embeddings, this product is essentially rotation in the complex plane, thus the same as the idea in this paper.

The authors should clarify and emphasize how their model could provide advantage over the ComplEx model, which is currently one of the SOTA. The authors should provide convincing theoretical arguments because many researches have shown that excessive hyper-parameter tuning and optimization techniques can change benchmark results a lot [2]. The authors also need to provide proof that the ComplEx model cannot model "composition" as in Table 2, given the two models are essentially similar.

Additionally, the comparison with TransE is ambiguous. The authors should make clear that the rotation is in the complex plane of each embedding vector element, thus different from rotation in the embedding space; and check that their arguments and analyses regarding TransE still stand.

About experiments, for fair comparisons, results should be reported on common and standard settings. An example practice could be seen in [3].

Ref:
[1] Trouillon, Theo, et al. Complex Embeddings for Simple Link Prediction. ICML 2016.
[2] Kadlec, Rudolf, Ondrej Bajgar, and Jan Kleindienst. "Knowledge base completion: Baselines strike back." arXiv preprint arXiv:1705.10744 (2017).
[3] Lacroix, Timothée, Nicolas Usunier, and Guillaume Obozinski. "Canonical Tensor Decomposition for Knowledge Base Completion." ICML 2018.

---

> ### Public Comment · (anonymous) · 2018-11-13
> **Composition v.s. modelling 1-to-N, N-to-1, N-to-N relations in Table 2**
>
> This paper argues that the advantage of the proposed method against ComplEx is its ability to model composition. While this is true, the disadvantage of the TransE-type model (which includes RotatE) is its inability to deal with 1-to-N, N-to-1, N-to-N relations. It seems to me that the composition and modeling of these complicated relations are intrinsically at odds with each other. The author should make this clear, especially in Table 2; ComplEX can handle 1-to-N, N-to-1, N-to-N relations, while RotatE cannot.

---

> > ### Author Response · Authors · 2018-11-28
> > **Please refer to our reply to Reviewer2**
> >
> > Thanks for such a good question! We have provided some theoretical analysis to show that the RotatE model can also somehow model the 1-to-N relations. Please refer to our response to Reviewer2.

---

> > > ### Public Comment · (anonymous) · 2018-11-29
> > > **Thanks for the answer!**
> > >
> > > Thanks for the great answer! It makes sense to me!
> > > Probably, when N is large (in 1-to-N relation), it is better for TransE-type model to down-weight the corresponding loss term, so that those N entities will not be forced to have very similar embeddings.
> > >
> > > Another related question: how does the training loss behave for your model? Does it perfectly fit the training set?

---

> > > > ### Author Response · Authors · 2019-02-17
> > > > **Yes, RotatE can perfectly fit the training set**
> > > >
> > > > Here is our results on FB15k training set:
> > > >
> > > > Task Prediction Head (MRR) Prediction Tail (MRR) MRR
> > > > Relation Category 1-to-1 1-to-N N-to-1 N-to-N 1-to-1 1-to-N N-to-1 N-to-N Overall
> > > > RotatE 0.998 1.000 0.969 0.999 0.998 0.961 1.000 0.999 0.995

---

> ### Author Response · Authors · 2018-11-26
> **Thanks for your comments!!**
>
> Thanks for your comments!! The difference between RotatE and ComplEx can be summarized as follows:
>
> (1)ComplEx belongs to the semantic matching model while RotatE belongs to the distance-based model. Most of existing knowledge graph embedding models can be roughly classified into two categories: Translational(Transformational) Distance Models and Semantic Matching Models [1]. The former measure the plausibility of a fact as a translation(transformation) between two entities, while the latter measure the plausibility of facts by matching latent semantics of entities and relations. RotatE and ComplEx are in different categories. Actually, we can find that the relation between ComplEx and RotatE is in analogy to the relation between TransF [2] and TransE, where the former can be regarded as a slack version of the latter.
>
> (2) As a result, the biggest difference between ComplEx and RotatE addressed in this paper is that, the RotatE model can infer the composition pattern of relations, while the ComplEx model cannot. A simple counterexample could illustrate this point.
>
> Let’s assume r1(x, y), r2(y, z) and r3(x, z) hold, and then according to ComplEx we have
>
> Re(<r1, x, \bar{y}>)  >  Re(<r1, x’, \bar{y’}>)
> Re(<r2, y, \bar{z}>)  >  Re(<r2, y’, \bar{z’}>)
> Re(<r3, x, \bar{z}>)  >  Re(<r3, x’, \bar{z’}>)
>
> where r1(x’, y’), r2(y’,z’) and r3(x’, z’) are negative triplets.
>
> From the above equations, we can find that the ComplEx model does not model a bijection mapping from h to t via relation r.  For example, let x=-1+i, y=1, z=1+i, r1=-1-0.8i, r2= 0.2+i, r3=-0.8-i, we have r1(x, y), r2(y, z) and r3(x, z) hold, because
>
> <r1, x, \bar{y}> = 1.8 -  0.2i
> <r2, y, \bar{z}> = 1.2 + 0.8i
> <r3, x, \bar{z}> = 2 - 1.6i
>
> However, r1 * r2 = 0.6 - 1.16i, r3= - 0.8 - i do not show the supposed pattern r1 \circ r2 = \alpha r3 here.
>
> As for the comparison with TransE, the rotation in the RotatE model is in the complex plane of each embedding vector element, as the same as TransE. This is different from the rotation is in the whole embedding space by matrix multiplication.
>
> “About experiments, for fair comparisons, results should be reported on common and standard settings, especially with and without new negative sampling method….”
>
> We have added the results of TransE and ComplEx with the new adversarial negative sampling technique on three datasets in Table 8.
>
> “The authors should also address how they estimate/or approximate the softmax in Equation 4 of negative sampling method to scale to large datasets, because it is very costly due to the normalization term. ...”
>
> p(h’_j , r, t’_j |{(h_i , r_i , t_i)}) is defined as the probability that we sample (h’_j , r, t’_j) from a sampled set {(h_i , r_i , t_i)}, so we calculate the softmax function only on the sampled triplets. This is very efficient.
>
> “ It's also not clear what $ f_r $ refers to in Equation 4.”
>
>  $f_r$ is the score function introduced in Table 1, which equals to $- d_r$.
>
> [1]  Knowledge Graph Embedding: A Survey of Approaches and Applications
> [2]  Knowledge graph embedding by flexible translation

---

> > ### Public Comment · (anonymous) · 2018-11-30
> > **The claim about composition pattern**
> >
> > Thanks for your response. However, I think your example of the ComplEx model missed the point. Moreover, it is not a proof that ComplEx cannot model composition. In fact, the example has reasoning error. I can always start from picking r1 \circ r2 != alpha r3 then picking x, y, z that satisfies <r1, x, \bar{y}>, <r2, y, \bar{z}>, and <r3, x, \bar{z}>. One example does not make a  proof.
> >
> > The point is, as you have many strong claims, I expect to see the proofs, either mathematical proof, or clear empirical evidences. For example, showing ComplEx fails miserably on synthetic data with composition pattern.
> >
> > Update: We should focus on main points. Please justify your claim about composition pattern. Thanks again.

---

### Public Comment · (anonymous) · 2018-12-16
**Great paper! Results of RotatE without self-adversarial training?**

This is a great paper with strong empirical performance!!

I suppose you have also tried RotatE without self-adversarial training. Was it still better than all the other baselines (without self-adversarial training)? Or is it the combination of RotatE and self-adversarial that is crucial?

I think it is also necessary to put extensive results of all the baselines with self-adversarial training on *ALL* the datasets. When proposing two complementary methods, it is crucial to clearly separate the contribution. To me, it is surprising that self-adversarial training alone can significantly boost the performance of all the methods, and the training strategy is already a great contribution.

---

> ### Public Comment · (anonymous) · 2018-12-21
> **Self-adversarial sampling in the literature**
>
> The "self-adversarial" sampling is not a new technique. It dated back to at least 2016, as used in training sentence embedding [1]. Using this technique usually helps, but sometimes it is tricky because of more false-negative. How exactly the author implemented this technique in their model to avoid false-negative is an interesting question.
>
> [1] Wieting, J., Bansal, M., Gimpel, K., & Livescu, K. (2015). Towards universal paraphrastic sentence embeddings. arXiv preprint arXiv:1511.08198. ICLR '16.

---

> > ### Public Comment · (anonymous) · 2018-12-25
> > **Then, RotatE without self-adversarial training is the real contribution.**
> >
> > I think the performance of RotatE without self-adversarial sampling should be extensively reported.
> > Also, there is no convincing explanation as to why self-adversarial training helps RotatE. If we do extensive hyper-parameter search on self-adversarial sampling, maybe ComplEx can be better. Self-adversarial sampling is really a complementary method that is not tailored to RotatE.

---

> > ### Public Comment · (anonymous) · 2018-12-25
> > **Do you think [1] is sampling?**
> >
> > I don't think the "SELECTING NEGATIVE EXAMPLES" process in [1] is sampling.
> >
> > '''
> > To select t1 and t2 in Eq. 1, we tune the choice between two approaches. The first, MAX, simply chooses the most similar phrase in some set of phrases (other than those in the given phrase pair). For simplicity and to reduce the number of tunable parameters, we use the mini-batch for this set, but it could be a separate set. Formally, MAX corresponds to choosing t1 for a given hx1, x2i as follows:
> > t1 = argmax cos(g(x1), g(t))
> > where Xb ⊆ X is the current mini-batch. That is, we want to choose a negative example ti that is similar to xi according to the current model parameters. The downside of this approach is that we may occasionally choose a phrase ti that is actually a true paraphrase of xi.
> > '''
> >
> > To me, self-adversarial sampling is like a self-adversarial variant of KBGAN [2] and seems more elegant than [1].
> >
> > [1] Wieting, J., Bansal, M., Gimpel, K., & Livescu, K. (2015). Towards universal paraphrastic sentence embeddings. arXiv preprint arXiv:1511.08198. ICLR '16.
> > [2] Liwei Cai, & William Yang Wang. (2017). KBGAN: Adversarial Learning for Knowledge Graph Embeddings. NAACL '18

---

> > > ### Public Comment · (anonymous) · 2019-01-02
> > > **Wieting et al. do self-adversarial negative sampling**
> > >
> > > Sampling and selecting examples are equivalent terms here.
> > >
> > > The method to choose $t_1$ is essentially "self-adversarial" negative sampling. Notice that by $t_1 = argmax cos(g(x_1), g(t))$, $t_1$ is the most difficult example regarding the model itself. Wieting et al. went even further with addressing the false-negative problem.
> > >
> > > The current work should focus on showing how they apply it effectively.

---

> > > > ### Public Comment · (anonymous) · 2019-01-03
> > > > **Then this is why negative examples in Wieting et al. are not effective**
> > > >
> > > > Because Wieting et al. only used the most difficult example, which is highly possible to be false-negative. In contrast, KBGAN and self-adversarial sampling do not seem to suffer from the false-negative problem.

---

> > > > > ### Public Comment · (anonymous) · 2019-01-13
> > > > > **Acknowledging and comparison with related work**
> > > > >
> > > > > I think you may have misread or misunderstood the above comment and the Wieting et al. paper. Wieting et al.'s method is self-adversarial negative sampling. It seems to me that their method is more advanced because they identified and addressed the false negative problem. I would be happy to see a constructive comparison.
> > > > >
> > > > > On the other hand, like other comments said, reported results are buried under many optimization techniques and tunings. As a fellow researcher I would like to see more direct results.

---

> > > > > > ### Public Comment · (anonymous) · 2019-01-13
> > > > > > **Recommending a paper**
> > > > > >
> > > > > > I would like to recommend the original GAN paper [1] to resolve your misunderstanding on adversarial fake sample generation. Sampling (equivalent to sample from a fake "distribution" here, not only the most misleading sample) is the key to the success in GAN.
> > > > > >
> > > > > > If the generator only generates the most misleading sample, it's easy to see that the  Global Optimality is not
> > > > > >
> > > > > > p_g = p_data.
> > > > > >
> > > > > > Because the optimal discriminator D now is
> > > > > >
> > > > > > D∗_G(x) = 0 if (p_g(x) >= p_g(*)) else 1
> > > > > >
> > > > > > [1] Goodfellow I, Pouget-Abadie J, Mirza M, Xu B, Warde-Farley D, Ozair S, Courville A, Bengio Y. Generative adversarial nets. NIPS 2014

---

### Public Comment · ~Apoorv_Umang_Saxena1 · 2019-05-28
**Why are these 2 numbers different?**

In table 7: self-adversarial, FB15K-237, H@10 is 0.465

In table 8: Transe, FB15K-237, H@10 is 0.531

Shouldn't they be same?

---

> ### Author Response · Authors · 2019-05-28
> **Their embedding dimensions are different**
>
> ````""
> We re-implement a 50-dimension TransE model with the margin-based ranking criterion that was used in (Cai & Wang, 2017), and evaluate its performance on FB15k-237, WN18RR and WN18 with self-adversarial negative sampling.
> ""

---

### Public Comment · ~Rajiv_Teja_Nagipogu1 · 2019-11-07
**Why are the dimensions of entity embedding and relation embedding different for RotatE in the code?**

I am relatively new to the KG embedding space, so forgive me if this is something trivial.

I ran the code specified in the paper, with the same parameters as the README. I can see that -de option is specified but not -dr which makes the dimensions of entity and relation embeddings different. However, you are assuming tail as the Hadamard product of these embeddings which require them to have same dimensions. What am I missing? Thanks. Good work!

---

> ### Author Response · Authors · 2019-11-07
> **This is an implementation for the modulus constraint for the relation embeddings**
>
> Hi Rajiv,
>
> This is an implementation choice for the modulus constraint for the relation embeddings. In this repository, we use these real-valued vectors to represent the phases of the relation embeddings, while use doubled real-valued vectors to represent complex-valued embeddings.
>
> A relevant discussion can be found at https://github.com/DeepGraphLearning/KnowledgeGraphEmbedding/issues/7

---

### Public Comment · (anonymous) · 2022-05-14
**more explanation for the Self-adversarial negative sampling step**

I think self-adversarial negative sampling is the same as KBGAN but instead of two models , it is just one model and specific way to draw the negative samples

as I think:

1- sample negative sampling from the set of all entities depend on the softmax function for all entities with adversarial temperature to adjust the sampling to be not like Uniform or argmax (is this right?)
2- take the negative sample with the highest probability from the previous function and then use in the loss function?

Also in the function why :    hi Numerator   and then hj in denominator  expfr(hi,ti)  / sum (expfr(hj,tj))

---

> ### Public Comment · (anonymous) · 2022-10-16
> **sample negative sampling from the set of all entities by argmax-loss is expensive**
>
> The idea you think may be the same with the following paper, which is slightly earlier than this paper.
> Lei, J., Ouyang, D., & Liu, Y. (2019). Adversarial Knowledge Representation Learning Without External Model. IEEE Access, 7, 3512-3524.
> https://ieeexplore.ieee.org/document/8599182
>
> A slightly difference between the paper self-adaptive and self-adv. is that fixed number of negative samples (negative samples n = 64/128/256/1024...) are exploited for one positive case instead of 1 negative case for one positive case from fixed number of random selected cases (Ns = 20).
> Uniformly negative sampling is cheap, but negative sampling from the set of all entities by argmax-loss is expensive. It may work but may be not necessary in training.

---

### Meta-Review · Area_Chair1 · 2018-12-17
**Interesting idea, solid results, good analysis**

**Confidence:** 4
**Recommendation:** Accept (Poster)

**Metareview:**

This paper proposes a knowledge graph completion approach that represents relations as rotations in a complex space; an idea that the reviewers found quite interesting and novel. The authors provide analysis to show how this model can capture symmetry/assymmetry, inversions, and composition. The authors also introduce a separate contribution of self-adversarial negative sampling, which, combined with complex rotational embeddings, obtains state of the art results on the benchmarks for this task.

The reviewers and the AC identified a number of potential weaknesses in the initial paper: (1) the evaluation only showed the final performance of the approach, and thus it was not clear how much benefit was obtained from adversarial sampling vs the scoring model, or further, how good the results would be for the baselines if the same sampling was used, (2) citation and comparison to a closely related approach (TorusE), and (3) a number of presentation issues early on in the paper.

The reviewers appreciated the author's comments and the revision, which addressed all of the concerns by including (1) additional experiments to performance with and without self-adversarial sampling, and comparisons to TorusE, (2) improved presentation.

With the revision, the reviewers agreed that this is a worthy paper to include in the conference.

---

> ### Public Comment · (anonymous) · 2018-12-21
> **Thanks for your hard work, and a few comments**
>
> I am concerned with the style of presentation in this paper.
>
> 1. "rotation in complex plane": in knowledge graph embedding community, the ComplEx model [1] is very well established. It involves complex number product, which is "rotation in complex plane". The authors failed to compare to existing work.
>
> 2. "self-adversarial negative sampling": this technique was used at least in 2016 to train sentence embedding [2]. The authors also failed to compare to existing work.
>
> 3. Reporting of result: for fair comparison to future work (and also past work), the paper should include results of RotatE on standard setting.
>
> The result is worthy, nevertheless the writing's getting on my nerve. This is only my ranting, but that's what gets to people working closely with this topic. This is in a comment to AC because the conference can choose which practice to endorse. I urge the authors to rewrite in a more straightforward and fair style.
>
> [1] Trouillon, Théo, et al. "Complex embeddings for simple link prediction." ICML '16.
> [2] Wieting, John, et al. "Towards universal paraphrastic sentence embeddings." ICLR '16.